# Surface hydrophobization of zeolite enables mass transfer matching in gas-liquid-solid three-phase hydrogenation under ambient pressure

Shuai Wang[1], Riming Hu[2], Jianyu Ren[3], Yipin Lv[1], Lianghao Song[4], Huaiqing Zhao[1], Xuchuan Jiang[2], Daowei Gao [1] ✉ & Guozhu Chen[1] ✉

Attaining high hydrogenation performance under mild conditions, especially at ambient pressure, remains a considerable challenge due to the difficulty in achieving efficient mass transfer at the gas-liquid-solid three-phase interface. Here, we present a zeolite nanoreactor with joint gas-solid-liquid interfaces for boosting $H_2$ gas and substrates to involve reactions. Specifically, the Pt active sites are encapsulated within zeolite crystals, followed by modifying the external zeolite surface with organosilanes. The silane sheath with aerophilic/hydrophobic properties can promote the diffusion of $H_2$ and the mass transfer of reactant/product molecules. In aqueous solutions, the gaseous $H_2$ molecules can rapidly diffuse into the zeolite channels, thereby augmenting $H_2$ concentration surround Pt sites. Simultaneously, the silane sheath with lipophilicity nature promotes the enrichment of the aldehydes/ketones on the catalyst and facilitates the hydrophilia products of alcohol rediffusion back to the aqueous phase. By modifying the wettability of the catalyst, the hydrogenation of aldehydes/ketones can be operated in water at ambient $H_2$ pressure, resulting in a noteworthy turnover frequency up to 92.3 $h^{-1}$ and a 4.3-fold increase in reaction rate compared to the unmodified catalyst.

Liquid-phase hydrogenation of aldehydes/ketones to corresponding alcohols, with cost-effective and eco-friendly features, constitutes a promising route to produce vital organic intermediates for pharmaceuticals, fragrances, preservatives, etc[1,2]. Among the present approaches, heterogeneous catalytic hydrogenation is a suitable candidate because the catalyst is easily recovery and the ideal byproduct is just renewable $H_2O$. Currently, efforts in this route have been extensively investigated[3–6]. However, conventional hydrogenation catalysts generally necessitate stringent conditions to attain high reaction rate[7,8]. Therefore, the development of efficient catalytic system for the hydrogenation of ketones/aldehydes under mild and environmentally

friendly conditions, particularly under ambient pressure, is highly desired.

The mass transfer in hydrogenation, an important influencing factor on catalytic performance, has long been neglected. With respect to hydrogenation on gas-liquid-solid three phase, the dissolution of gaseous $H_2$ molecules in solvent as well as the diffusion to metal sites for activation are both essential but important steps during the catalytic process. However, at ambient pressure, owing to the quite low solubility of gaseous $H_2$ in solvent, the efficiency of reactions is significantly suppressed[8–10]. Moreover, the substrate molecules cannot spontaneously enrich on the catalyst surface due to the liquid-solid

[1]School of Chemistry and Chemical Engineering, University of Jinan, Jinan 250022, PR China. [2]Institute for Smart Materials & Engineering, University of Jinan, Jinan 250022, PR China. [3]Department of Chemistry and Biochemistry, University of California San Diego, La Jolla, CA 92093, USA. [4]Department of Chemistry, Sungkyunkwan University, Suwon 16419, Korea. ✉e-mail: chm_gaodw@ujn.edu.cn; chm_chengz@ujn.edu.cn

mass transfer limitation. In this regard, a method for solving the gaseous $H_2$-deficit problem and adjusting the substrate diffusion for the specific catalytic systems is highly anticipated.

The wettability has emerged to be sensitive to the performances of heterogeneous catalysts by constructing hydrophobic or hydrophilic surface, because it greatly influences the diffusion of reactant and product molecules[11–18]. Furthermore, hydrophilic or hydrophobic catalyst surfaces have particular affinity for the corresponding substrates, with beneficial consequences for catalytic conversion[19–21]. Inspired by this, the issue of mass transfer mismatch in the gas-liquid-solid three-phase hydrogenation can also be tackled through the wettability modification utilizing an aerophilic/hydrophobic surface. Specifically, superhydrophobic surface in air would be underwater superaerophilic, thus facilitating rapid diffusion of gaseous $H_2$ on the catalyst surface. Meanwhile, hydrophobic surface is usually lipophilic, permitting the enrichment of lipophilic substrates on catalyst. Moreover, such surface also can promote swift diffusion of hydrophilic products, which can facilitate the separation of molecules from the reaction system. However, controlling gas-liquid-solid mass transfer matching is rarely studied, which might be due to the limitation of methodology for tailoring molecular diffusion against liquid and gas phases simultaneously.

Herein, we report a nanoreactor with joint gas-solid-liquid interfaces and controlled wettability for boosting $H_2$ gas and substrates to involve reactions (Fig. 1). Specifically, a micro-mesoporous zeolite is used for fixing Pt active sites, followed by hydrophobic modification of the zeolite surface with organosilanes. Because of the hydrophobic/aerophilic property of the silane sheath, $H_2$ molecules can transport into zeolite channels with almost no resistance and reach the Pt sites directly from the gas phase to catalyst, significantly improving the concentration of $H_2$ for the reaction. Meanwhile, the lipophilic substrate molecules are affinitive to the silane sheath, which promotes the enrichment of substrate molecules around the Pt sites. Further, after hydrogenation, the hydrophilic alcohol product can promptly diffuse into the aqueous solvent. As a result, the reaction efficiency of benzaldehyde hydrogenation is enhanced up to 4.3-fold compared with the traditional catalyst. Such strategy can be adapted to the hydrogenation of a broad substrate scope of aldehydes/ketones. This work represents an advancement towards the rational design of mass transfer-matched hydrogenation catalysts.

## Results and discussion
### Catalyst synthesis and structure characterizations
To prepare the catalyst, the Pt species were encapsulated within hierarchical titanium silicalite-1 crystals (Pt@HieTS-1) using a ligand-protected hydrothermal approach. Chloroplatinic acid hexahydrate ($H_2PtCl_6 \cdot 6H_2O$) and (3-Mercaptopropyl)trimethoxysilane were utilized as the Pt precursor and the corresponding coordination ligand, respectively[22]. Subsequently, the Pt@HieTS-1 catalyst was obtained through crystallization, filtration, and calcination. Then, the external-surface hydrophobization of zeolite is obtained by modification with organosilanes. The catalysts are denoted as Pt@HieTS-1-$C_x$, where the @ symbol is related to the encapsulation of the Pt active sites within the hierarchical HieTS-1 zeolite and $C_x$ represents the organic substituent of the silane. Specifically, trimethoxy(propyl)silane, trimethoxyphenylsilane and hexadecyltrimethoxysilane are labeled as -$C_3$, -$C_6$, and -$C_{16}$, respectively. The organic groups ($C_x$) that form the hydrophobic coating were introduced onto the zeolite crystals via post-silylation[12].

As shown in Fig. 2, high-resolution transmission electron microscopy (HRTEM) reveals that the Pt@HieTS-1 catalyst consists of regularly self-pillared zeolite crystals, with an average size of approximately 180 nm (Fig. 2a and Supplementary Figs. 1, 2). The Pt@HieTS-1 catalyst displays the characteristic topological structure of MFI (Fig. 3a) with a typical micro-mesoporous structure (Fig. 3b)[23–26].

Cs-corrected STEM images (Fig. 2d, e) further evidence that Pt@HieTS-1 exhibits a superior crystalline structure, explicit crystal lattice, and an unimpaired MFI-type framework structure along the [010] orientation. Within the zeolite, nanoparticles (NPs) with lattice spacings of 0.23 nm and 0.20 nm (Fig. 2b, c) are clearly observed, which are unmistakably attributed to Pt (111) and Pt (200), correspondingly. In addition to the Pt NPs, a considerable number of Pt single atoms were also confirmed by the Cs-corrected HAADF STEM images of Pt@HieTS-1 (Fig. 2d). Cs-corrected iDPC STEM images disclose that the Pt single atoms are located in the 5- and 6-membered rings (MR) of the MFI zeolite framework (Fig. 2e). This configuration was optimized by density functional theory (DFT) calculations (Supplementary Fig. 5). The enlarged images of three typical local structures and corresponding binding energies were listed in Fig. 2e. Energy dispersion spectroscopy (EDS) element mapping results demonstrate that Pt species distribution is homogeneous within a single Pt@HieTS-1 nanocrystal (Fig. 2g).

The oxidation states of Pt in zeolites were investigated using X-ray photoelectron spectroscopy (XPS). Figure 3c illustrates that both metallic and oxidized Pt species could be differentiated in the Pt $4f$ XPS spectra[27,28]. The oxidation states and microenvironment of Pt species in HieTS-1 were also determined via X-ray absorption spectroscopy (XAS). The X-ray absorption near-edge structure (XANES) spectra at Pt $L_3$-edge for Pt@HieTS-1 (Fig. 3d) verified the formation of partial positively charged $Pt^{\delta+}$ species, as the Pt white-line intensity for Pt@HieTS-1 was positioned between those of $PtO_2$ and Pt foil, signifying the presence of electronic interaction between Pt species and the O atoms in the 5- and 6-MR of MFI[29–31]. The Fourier-transformed extended X-ray absorption fine spectroscopy (FT-EXAFS) analysis of Pt@HieTS-1 (Fig. 3e) displays a notable peak at ~1.6 Å attributable to the first shell of the Pt-O path as well as a weak peak at ~2.6 Å correlated with the second shell of the Pt-Pt path. Furthermore, the wavelet-transformed (WT) EXAFS oscillations of Pt@HieTS-1 also validate that the Pt@HieTS-1 possesses both the characteristic of Pt-O path (~6.0 Å$^{-1}$ in $k$ space and ~1.6 Å in $R$ space) and Pt-Pt path (~12.0 Å$^{-1}$ in $k$ space and ~2.6 Å in $R$ space) (Fig. 3j–i). Taken together, it can be inferred that the Pt species in HieTS-1 zeolites primarily exist as single Pt atoms bonded to the O atoms, along with a minor quantity of Pt NPs.

The successful appending of silane alkyl groups onto the zeolite crystals was verified via Fourier transform infrared (FTIR) spectroscopy (Fig. 4a). The Pt@HieTS-1-$C_3$ gives additional IR peaks at 2933, 2851, and 1387 cm$^{-1}$, which correspond to the stretching vibration of -$CH_3$ and -$CH_2$- groups, as well as the bending vibration of -$CH_3$, respectively[32]. Likewise, the presence of additional IR signals on the Pt@HieTS-1-$C_6$ and Pt@TS-$C_{16}$ (Supplementary Table 3) confirms the successful grafting of phenylsilane and hexadecylsilane, respectively, onto the zeolite surface. Moreover, XRD, $N_2$ adsorption-desorption isotherm and Pt $4f$ XPS analyses demonstrate that such modification had no significant impact on the pore structure of the zeolite or the chemical state of the Pt active sites (Fig. 3a–c).

Subsequently, the surface wettability of the catalysts was investigated. As depicted in Fig. 4b, Pt@HieTS-1 demonstrated hydrophilic characteristic with a water contact angle of 10.6° in air, which is attributed to the generation of the secondary porosity on the hierarchical zeolite, as this is expected to be covered by silanol groups[33–35]. As expected, the introduction of alkyl groups onto the zeolite surface led to a gradual increase in its hydrophobicity. The water contact angles of Pt@HieTS-1-$C_3$, Pt@HieTS-1-$C_6$, and Pt@HieTS-1-$C_{16}$ were measured to be 98.9°, 105.2°, and 109.3°, respectively. Therefore, tailoring the zeolite surface by this method can easily construct the desired hydrophobic structure. Since the water contact angle test only provides a macroscopic view of the hydrophobicity, confocal laser scanning microscopy (CLSM) was used to study the hydrophobicity at the micro-nanoscale. Water-soluble fluorescent dye (Fluorescein Sodium) was selected as the fluorescent probe molecule. In this regard, fluorescent dye can only lighten the area where water is present.

Figure 4c illustrates the schematic diagram and CLSM images of Pt@HieTS-1 and Pt@HieTS-1-C$_3$. The fluorescent dye illustrates the entire body of Pt@HieTS-1, while a feeble fluorescence signal is evident on the Pt@HieTS-1-C$_3$, indicating the incapability of water to infiltrate into the micro-mesoporous channels of Pt@HieTS-1-C$_3$.

A surface that is superhydrophobic in air would exhibit super-aerophilic properties when submerged in water. We conducted further investigations on the aerophilic properties of the catalysts. As shown in Fig. 4e, f, the adhesion behaviors of H$_2$ bubbles on the samples under water were examined. Observation reveals that Pt@HieTS-1

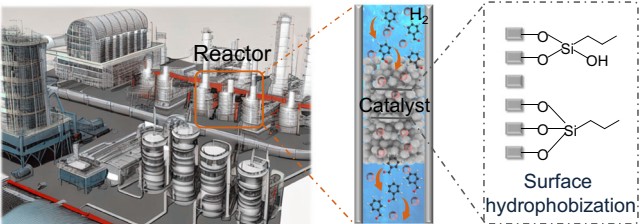

Efficient mass transfer in gas-liquid-solid three-phase hydrogenation

**Fig. 1 | Schematic representation of zeolite nanoreactor catalyst in gas-solid-liquid three phase hydrogenation.**

maintained the pinning state of the H$_2$ bubble for a duration of 6 s under water, indicating the aerophobic nature of this surface. Consequently, the spreading of H$_2$ gas across the surface of the Pt@HieTS-1 was impeded. On the contrary, the Pt@HieTS-1-C$_3$, Pt@HieTS-1-C$_6$, and Pt@HieTS-1-C$_{16}$ displayed superaerophilic properties with respect to the underwater H$_2$ bubbles. Bubbles exhibited bursting behavior and were able to diffuse completely in only 30 ms, indicating that H$_2$ gas can rapidly diffuse into the channels of the modified zeolite.

Besides the transportation of H$_2$ gas, another equally important issue is to elucidate the diffusion mechanism of substrates to the active sites inside the organosilane-modified zeolite. The whole Pt@HieTS-1 catalyst is susceptible to wetting by water, enabling benzaldehyde molecules to access the surface of Pt active sites in aqueous solution. However, on the Pt@HieTS-1-C$_x$, the micro-mesoporous channels are filled with H$_2$ and are impermeable to water. Therefore, an alternative pathway must be traversed by the benzaldehyde molecules on Pt@HieTS-1-C$_x$. The adsorption of benzaldehyde on the catalysts is confirmed by in situ diffuse reflectance infrared Fourier transform spectroscopy (DRIFTS) spectra (Fig. 4d)[36–39]. It is worth noting that the adsorption strength of benzaldehyde on Pt@HieTS-1-C$_3$ is significantly enhanced compared with that on Pt@HieTS-1, because the lipotropy of silane sheath promotes benzaldehyde enrichment on the catalyst surface. With the carbon chain length of organosilane precursor extending, it is slightly difficult for benzaldehyde to enter the pores of

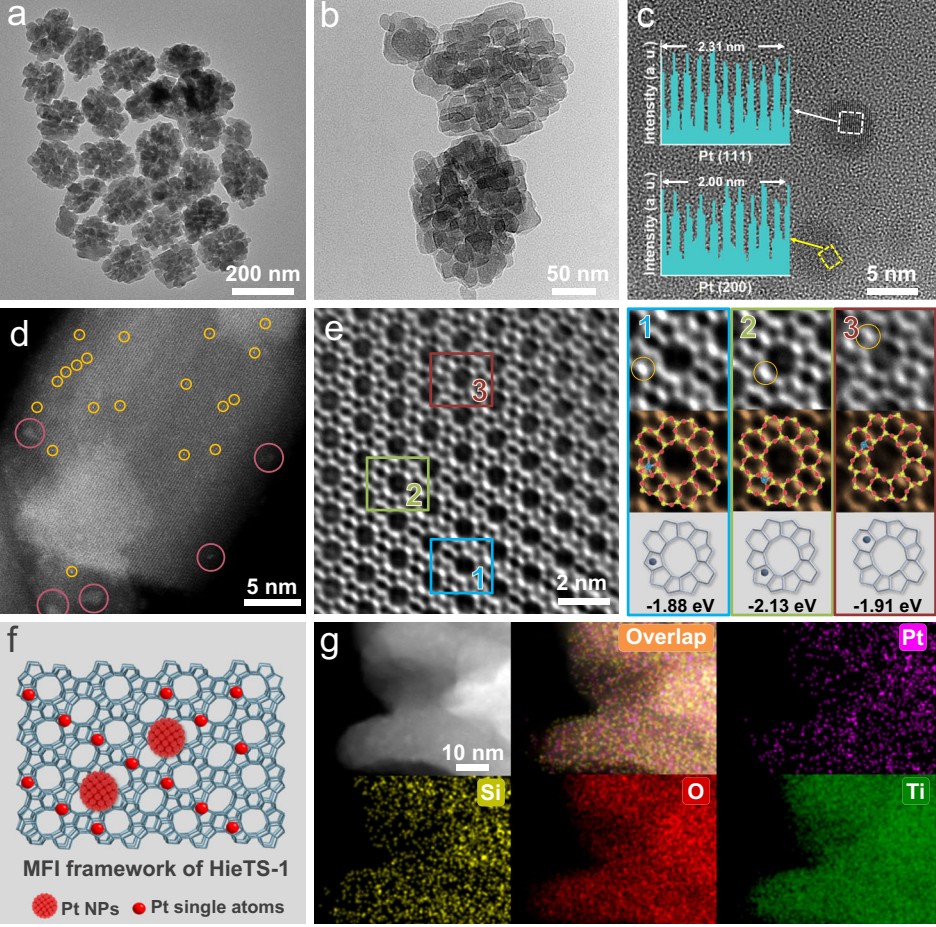

**Fig. 2 | Morphological and structural characterization. a, b** TEM images of Pt@HieTS-1 zeolite. **c** HRTEM image of Pt@HieTS-1 zeolite. The Pt NPs are highlighted within the dotted square, and the corresponding lattice is shown in illustration. **d** Cs-corrected HAADF STEM image of Pt@HieTS-1 zeolite. The Pt NPs and Pt single atoms are highlighted within the yellow circle and red circle, respectively. **e** High-magnification Cs-corrected iDPC STEM images of Pt@HieTS-1 zeolite. Zoomed-in areas of 1, 2, and 3 in (**e**) are the location of atomically dispersed Pt species in HieTS-1 zeolite framework and corresponding binding energy. Si (yellow), O (red) and Pt (blue). **f** Schematic diagram of the HieTS-1 zeolite framework and the location of Pt species. **g** EDS element mapping of Pt@HieTS-1 zeolite.

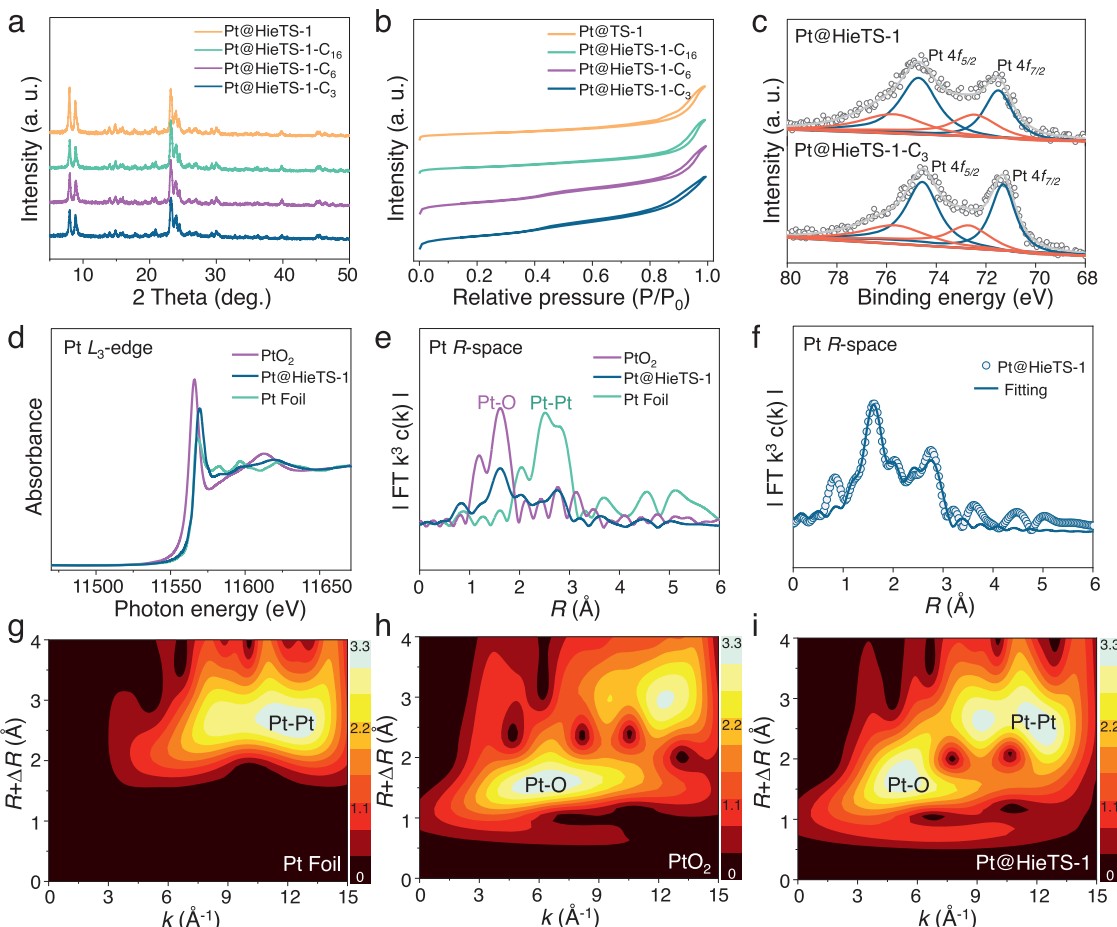

**Fig. 3 | The structure and chemical state investigation. a** XRD patterns, (**b**) $N_2$ adsorption-desorption isotherms and (**c**) Pt 4$f$ XPS of catalysts. Original data (gray circle), fitting (gray line) $Pt^{2+}$ (red line) and $Pt^0$ (blue line). **d** Pt $L_3$-edge XANES spectra of Pt@HieTS-1, $PtO_2$, and Pt foil. **e** FT EXAFS spectra of Pt@HieTS-1, $PtO_2$ and Pt foil. **f** FT EXAFS fitting spectrum of Pt@ HieTS-1 at $R$ space. WT EXAFS spectra of (**g**) Pt foil, (**h**) $PtO_2$ and (**i**) Pt@HieTS-1.

zeolite because of limited space, but it can still be adsorbed by the Pt active sites inside the zeolite. Further, the benzaldehyde contact angle test results provide further evidence of this, as it is capable of wetting Pt@HieTS-1-$C_x$ (Supplementary Fig. 8). These results suggest that the silane sheath can greatly promote the enrichment of benzaldehyde on the catalyst and accelerate the mass transfer of the substrate to the active site in aqueous phase. In addition, due to the lipophilicity of silane sheath and the higher solubility of alcohols in water, the diffusion of the resulting alcohol product into the solvent is accelerated (Supplementary Fig. 21).

### Catalytic performance toward the aldehydes/ketone hydrogenation

To evaluate the hydrogenation performance of the organosilane-modified zeolite catalysts, aldehydes/ketone hydrogenation was carried out. As depicted in Fig. 5a, the benzaldehyde hydrogenation performance revealed that the Pt@HieTS-1-$C_x$ catalysts displayed good activity even at a temperature as low as 50 °C under 1 atm $H_2$ pressure (Supplementary Table 4). Notably, the substrate was completely converted within 2.0 h with about 100% benzyl alcohol yield on Pt@HieTS-1-$C_3$, and the carbon balance is up to 99% (Supplementary Table 5). Furthermore, the reaction kinetics were thoroughly investigated[40–42]. The reactivity of Pt@HieTS-1-$C_3$ is 4.3 times superior to that of Pt@HieTS-1 (Fig. 5b, c). The turnover frequency (TOF) was calculated to be 92.3 h$^{-1}$ for Pt@HieTS-1-$C_3$ (Fig. 5d), representing a 3.2 times increase over Pt@HieTS-1. Similar enhancements in hydrogenation performance were also observed for other

aldehyde and ketone substrates (Fig. 5g). Further increasing the carbon number of the modified alkyl leads to lower enhancement effect, most likely due to the blockage of benzaldehyde molecule diffusion caused by the long carbon-chain alkyl. Collectively, these results indicate that these catalysts with superaerophilic surface exhibit an elevated catalytic efficiency as opposed to the unaltered zeolite. In addition, Pt@HieTS-1-$C_3$ also boasts exceptional recyclability and stability. The HieTS-1-$C_3$ catalyst could be reused at least six times without activity loss in the hydrogenation of benzaldehyde (Supplementary Figs. 9–11).

Considering other reaction conditions remain constant, the difference in reaction kinetics is attributed to the distinction in their local reaction environments. To corroborate the effect of this local hydrogen-rich and substrate-rich environment, the reaction solvent was transitioned from water to ethanol, and the reactivity of Pt@HieTS-1-$C_3$ and Pt@HieTS-1 was found to be nearly identical (Supplementary Fig. 12). It is because, under this condition, ethanol can wet both Pt@HieTS-1-$C_3$ and Pt@HieTS-1 (Supplementary Fig. 13), thereby causing no differences in microenvironment between Pt@HieTS-1-$C_3$ and Pt@HieTS-1. With regards to the unmodified Pt@HieTS-1, the micro-mesoporous channels are filled with water. The $H_2$ supply solely by the dissolved $H_2$ in aqueous phase, ultimately leading to a considerably low $H_2$ concentration in the vicinity of Pt active sites. In contrast, the micro-mesoporous channels of Pt@HieTS-1-$C_x$ are permeated with gaseous $H_2$, leading to a substantially higher $H_2$ and substrate concentration surround Pt active sites, accelerating the hydrogenation reaction rate.

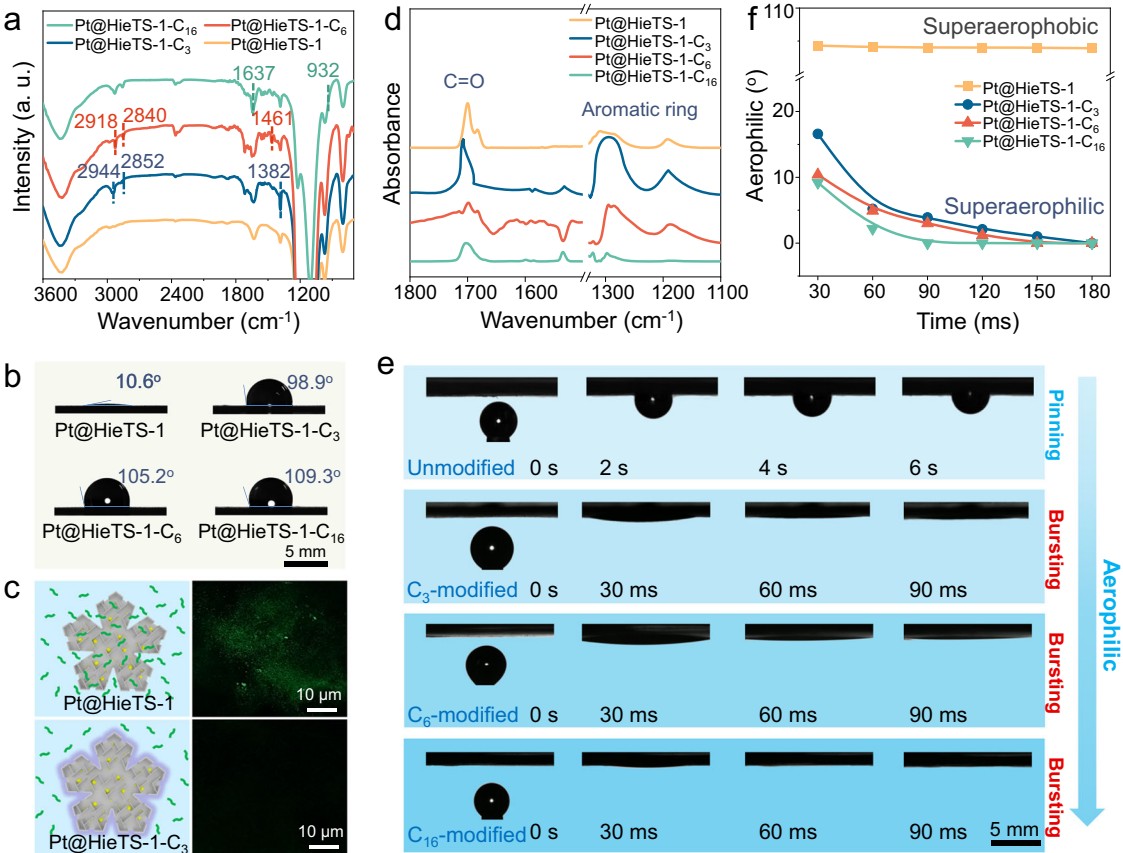

Fig. 4 | The wettability and adsorption capacity investigation. a IR spectra and (b) contact angles of water with Pt@HieTS-1, Pt@HieTS-1-C$_3$, Pt@HieTS-1-C$_6$ and Pt@HieTS-1-C$_{16}$. c Schematic diagram of Pt@HieTS-1 and Pt@HieTS-1-C$_3$ immersed in water containing fluorescent dye and CLSM images of Pt@HieTS-1 and Pt@HieTS-1-C$_3$. d In situ DRIFTS spectra of benzaldehyde adsorbed on Pt@HieTS-1, Pt@HieTS-1-C$_3$, Pt@HieTS-1-C$_6$ and Pt@HieTS-1-C$_{16}$. e The contact angles of H$_2$ gas-bubble with catalysts under water. f The relationship between H$_2$ gas-bubble contact angles and time over different catalysts, along with the Bezier-spline fitting.

The kinetic isotope effect observed in the H$_2$/D$_2$ system underscores the pivotal function of hydrogen dissociation in the hydrogenation of benzaldehyde (Fig. 5e and Supplementary Fig. 14). Therefore, promoting the hydrogen dissociation step can enhance the overall reactivity of the catalyst. For conventional catalysts in hydrogenation reactions, it is often imperative to apply a certain H$_2$ pressure to augment the H$_2$ concentration in the solution. The findings imply that catalysts possessing superaerophilic surfaces can elevate the H$_2$ concentration surrounding active sites, accelerating the reaction even under ambient pressure. These findings imply that high H$_2$ pressure may not be necessary for hydrogenation reactions when utilizing a catalyst with a superaerophilic surface. To further validate this notion, another reaction was conducted with Pt@HieTS-1 as the catalyst while raising the H$_2$ pressure to 1 MPa (Fig. 5f). The result revealed that Pt@HieTS-1 exhibited a comparable reaction rate under elevated H$_2$ pressure to that of Pt@HieTS-1-C$_3$ under ambient pressure. Furthermore, the catalytic activities of Pt@HieTS-1-C$_3$ and Pt@HieTS-1 were both improved after the increase of H$_2$ pressure, primarily attributed to the heightened solubility of H$_2$ (Supplementary Fig. 15). In particular, under identical H$_2$ pressure, the catalytic activity of Pt@HieTS-1-C$_3$ surpasses that of Pt@HieTS-1 significantly. This can be attributed to the unique hydrophobicity of the silane sheath in Pt@HieTS-1-C$_3$, which promotes the mass transfer of hydrophobic substrates to the active sites and facilitates the diffusion of hydrophilic products from the catalyst into the aqueous solvent. The temperature-dependent sensitivity of surface wettability was also investigated. As shown in Supplementary Fig. 16, within the temperature range of 50–110 °C, the catalytic performance of Pt@HieTS-1-C$_3$ catalyst markedly surpasses that of unmodified catalyst.

## Mechanism investigations of hydrogenation over Pt@HieTS-1-C$_x$

To gain deeper insight into the reaction mechanism, an isotope tracer method was performed to monitor the reaction process. When the hydrogenation reaction is carried out in an aqueous solvent, the H$_2$O involved H-exchange could serve as a significant pathway, and coexists with the direct H$_2$ dissociative hydrogenation pathway (Supplementary Fig. 17)[43,44]. Here, the H$_2$O solvent was replaced with D$_2$O for hydrogenation of benzaldehyde. Since the micro-mesoporous environment of Pt@HieTS-1 is full of water, a hydrogenation pathway of H-D exchange on Pt active sites is the dominant, leading to the appearance of a substantial number of deuterated benzyl alcohol products in the MS signal. As shown in Fig. 6b, the ratio of peak intensity at $m/z$ = M + 1 (where M represents molecular weight) to that at $m/z$ = M in the MS spectrum was high, exceeding 0.97. In contrast, the MS spectrum of Pt@HieTS-1-C$_3$ showed that this ratio exhibits a non-obvious change when compared to that of standard benzyl alcohol molecules (Fig. 6a, c). This implies that no active hydrogen species exchanged with D$_2$O at Pt active sites in the case of Pt@HieTS-1-C$_3$. The aforementioned findings unmistakably demonstrate that the distinct reaction pathways can be primarily attributed to the hydrogen-laden microenvironment of zeolite. The photographs of the WO$_3$ color change experiment (Fig. 6d) more intuitively confirm that the WO$_3$ discoloration is faster on Pt@HieTS-1-C$_x$. It can be attributed to the

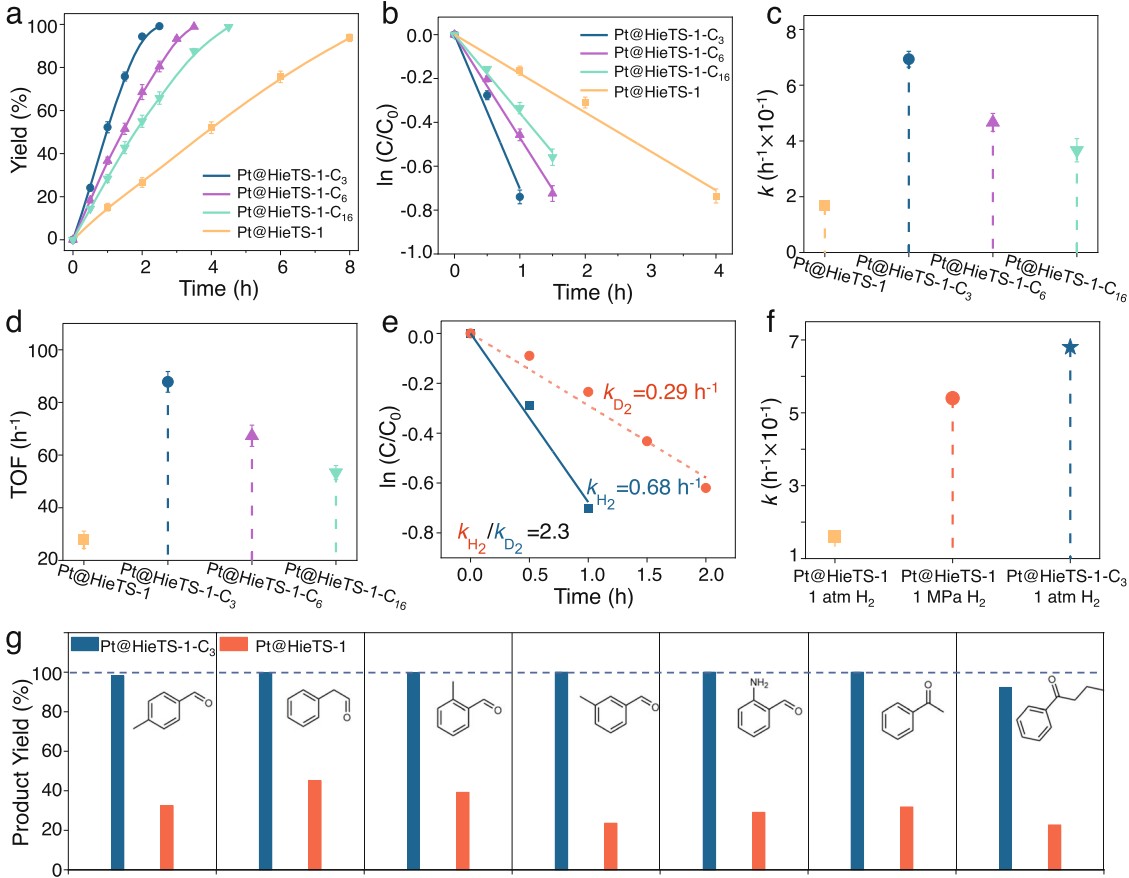

**Fig. 5 | Catalytic performances and kinetic studies. a** The evolution of benzaldehyde conversion with the reaction time on various catalysts, along with the Bezier-spline fitting. Reaction conditions: benzaldehyde (0.47 mmol), Pt dosage ($2.1 \times 10^{-3}$ mmol), water (5 mL), $H_2$ (1 atm), 50 °C. **b** The kinetic plots of the catalysts, along with the linear fitting. **c** The corresponding rate constant value of various catalysts. **d** The TOF values of the catalysts, calculated at a low conversion of benzaldehyde (20−30%). **e** Kinetic isotope effect of $H_2/D_2$ on benzaldehyde hydrogenation. **f** The rate constant value of Pt@HieTS-1 under $H_2$ pressure of 1 atm and 1 MPa. **g** The performance of Pt@HieTS-1-$C_3$ and Pt@HieTS-1 catalysts toward different substrates. For all graphs, data points indicate the group means and error bars represent the group standard deviation.

hydrogen-rich environment present within the zeolite, which facilitates the process of hydrogen spillover[45–47].

To confirm the above mechanism, a molecular dynamic (MD) simulation was employed to investigate the penetration behavior of the benzaldehyde and $H_2$ molecules across the TS-1 and TS-1-$C_3$ frameworks and to obtain a fundamental understanding of the molecular movement on hydrophobic and hydrophilic interfaces. In the initial state, 400 molecules of benzaldehyde (or $H_2$) molecules and 2640 molecules of water molecules (solvent) were located at the left of TS-1 and TS-1-$C_3$ with the distance about 0.5 nm (Supplementary Figs. 18–20). The simulation snapshots of MD are shown in Fig. 7. The mobility of benzaldehyde and $H_2$ diffusion into TS-1 or TS-1-$C_3$ is evaluated by mean-squared displacement (MSD). As expected, the mobility of benzaldehyde molecules into TS-1-$C_3$ is much higher than that of TS-1, because benzaldehyde molecules diffuse easier on a lipophilic surface than on a hydrophilic one (Fig. 7a, b). This result reveal that the diffusion rate of benzaldehyde into TS-1-$C_3$ is much faster than that of TS-1, consistent with the results of adsorption experiments (Supplementary Fig. 21). Additionally, the impact of superaerophilic surface on $H_2$ diffusion was also investigated to unravel the microscopic origin of hydrogen-rich environment within TS-1-$C_3$ (Fig. 7c, d). MD simulation reveals that the $H_2$ molecules diffuse more easily through the pore apertures of TS-1-$C_3$ than TS-1, in agreement with the results of $WO_3$ color change experiment. As depicted in Supplementary Table 5, the penetration ratios of benzaldehyde and $H_2$ molecules in the TS-1-$C_3$ framework are approximately

threefold higher than those in TS-1. This comparison further substantiates that benzaldehyde and $H_2$ molecules manifest a heightened propensity for diffusion into the zeolite channels within the TS-1-$C_3$ framework.

Based on the above analyses, two reasons can be accounted for the high-performance of Pt@HieTS-1-$C_3$ towards hydrogenation of aldehydes/ketones under ambient pressure (Fig. 6e). The first one is the hydrogen-rich microenvironment in channels of Pt@HieTS−1-$C_3$. After modifying with silane sheath on the zeolite surface, the micro-mesoporous channels of Pt@HieTS−1-$C_3$ develop into superaerophilic. This structural configuration is beneficial for the storage of $H_2$ molecules within zeolite channels. During the reaction, the concentration of $H_2$ confined in Pt@HieTS-1-$C_3$ is significantly higher than that in solution. Furthermore, with the assistance of stirring, Pt@HieTS-1-$C_3$ catalyst can reach the gas-liquid interface constantly to safely replenish $H_2$ from the upper atmosphere when $H_2$ within channels is consumed during the reaction. In contrast, for Pt@HieTS-1, since the zeolite channels are filled with water, the $H_2$ concentration around the Pt active sites is low, due to the low solubility of $H_2$ in water at ambient pressure (0.81 mM at 25 °C) (Fig. 6e). Obviously, this method obviously overcomes the gas-deficit problem in hydrogenation under ambient pressure. The second point is the efficient diffusion of substrates and products. The lipophilic substrate molecules are affinitive to the silane sheath, which promotes the enrichment of substrate molecules around the Pt sites. After hydrogenation, the hydrophilic alcohol product can promptly diffuse into the aqueous phase because of their

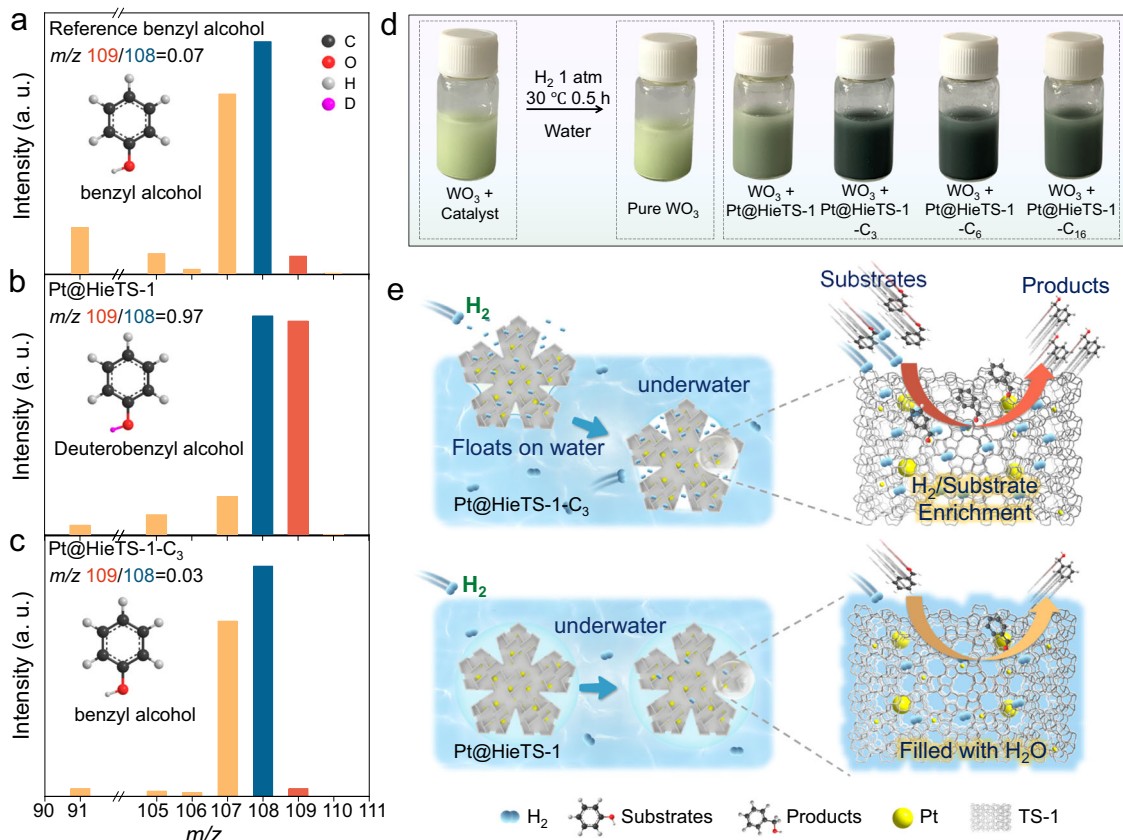

**Fig. 6 | Catalytic mechanism for hydrogenation. a** MS spectra of standard benzyl alcohol. Hydrogenation products in $D_2O$-labeling reaction experiments of (**b**) Pt@HieTS-1-$C_3$ and (**c**) Pt@HieTS-1. **d** Photographs of $WO_3$ mixed with or without catalysts treated under the reaction condition in the absence of benzaldehyde. **e** Schematic illustrations of the hydrogenation reaction occurring on Pt@HieTS-1-$C_3$ and Pt@HieTS-1.

similar polarity. This breaks the scaling relations between liquid-solid mass transfer, promoting the substrate enrichment around the active site and promoting the removal of the product from the catalytic system.

In summary, we have demonstrated the concept that a simple heterogeneous catalyst of Pt@HieTS-1-$C_x$ can perform efficient hydrogenation of aldehydes/ketones under ambient pressure by suitably engineering the wettability and pore microenvironment of the zeolite with organosilane modification. Due to the coverage of aerophilic/hydrophobic silane sheath, the Pt@HieTS-1-$C_x$ catalysts allow $H_2$ molecules to be filled and stored inside its porous channels. Moreover, the silane sheath has a strong affinity for the organic molecules, which promotes the substrates enrichment on the catalyst surface and facilitates the hydrophilia products rediffusion back to the aqueous phase. Thus, the efficient mass transfer of $H_2$ and substrate/product molecules are both greatly enhanced in compared to that of unmodified Pt@HieTS-1. This concept described herein offers a perspective toward the rational design of mass transfer-matched hydrogenation catalysts.

## Methods
### Catalyst synthesis
**Synthesis of hierarchical titanium silicalite-1 zeolite (Pt@HieTS-1).** The HieTS-1 zeolite fixed Pt species was synthesized via hydrothermal method using 40.0 g of tetraethyl orthosilicate, 2.6 g of tetrabutyl titanate, 30.0 g of ethanol and 46.7 g of TPAOH (25 wt%) as raw materials. The mixture is stirred in an ice bath for 1 h to blend thoroughly. Subsequently, the solution was transferred to a 70 °C water bath and stirred for 3 h, and then 48 g isopropyl alcohol was added and stirred for another 1 h to obtain solution I. The mixture of 0.1 g of

NaOH, 2.0 g water, and 0.2 g of 3-trimercaptopropyltrimethoxysilane was hydrolyzed by stirring for 0.5 h, then 6.16 ml of $H_2PtCl_6$ (100 mM) was added and stirred again for 0.5 h to obtain solution II. Solution II was slowly added into solution I. After further stirring for 0.5 h, the mixture was transferred into an autoclave to crystallize at 170 °C for 24 h. After filtrating, washing, drying and calcining (500 °C), the Pt@HieTS-1 was obtained.

**Synthesis of Pt@HieTS-1-$C_x$.** The Pt@HieTS-1-$C_x$ zeolite was synthesized by a postsilylation method. As a typical procedure for the synthesis of Pt@HieTS-1-$C_3$, 0.5 g of the Pt@HieTS-1 was dried at 150 °C under vacuum and then dispersed in 10 mL of anhydrous toluene by sonication at room temperature. Then, 0.5 g of trimethoxy(propyl)silane was dissolved in 20 mL of anhydrous toluene and the zeolite suspension was added to the solution under stirring. The mixture was stirred for 24 h at 500 rpm under room temperature. After filtrating, washing with ethanol, drying at 100 °C for overnight, the Pt@HieTS-1-$C_3$ was obtained. The Pt@HieTS-1-$C_6$ and Pt@HieTS-1-$C_{16}$ were synthesized according to similar procedures except for using trimethoxyphenylsilane (0.7 g) and hexadecyltrimethoxysilane (0.4 g) as precursors.

### Characterization
The structure of the catalysts was detected by transmission electron microscopy (TEM, JEM 2100), scanning electron microscopy (SEM, SU8010), and high-resolution TEM (FEI Tecnai G2 F20). Spherical aberration-corrected (Cs-corrected) scanning transmission electron microscopy (STEM) data were acquired on a JEOL GrandARM 300 instrument equipped with a double corrector. The X-ray powder diffraction (XRD) patterns were collected at a Bruker D8 advance powder

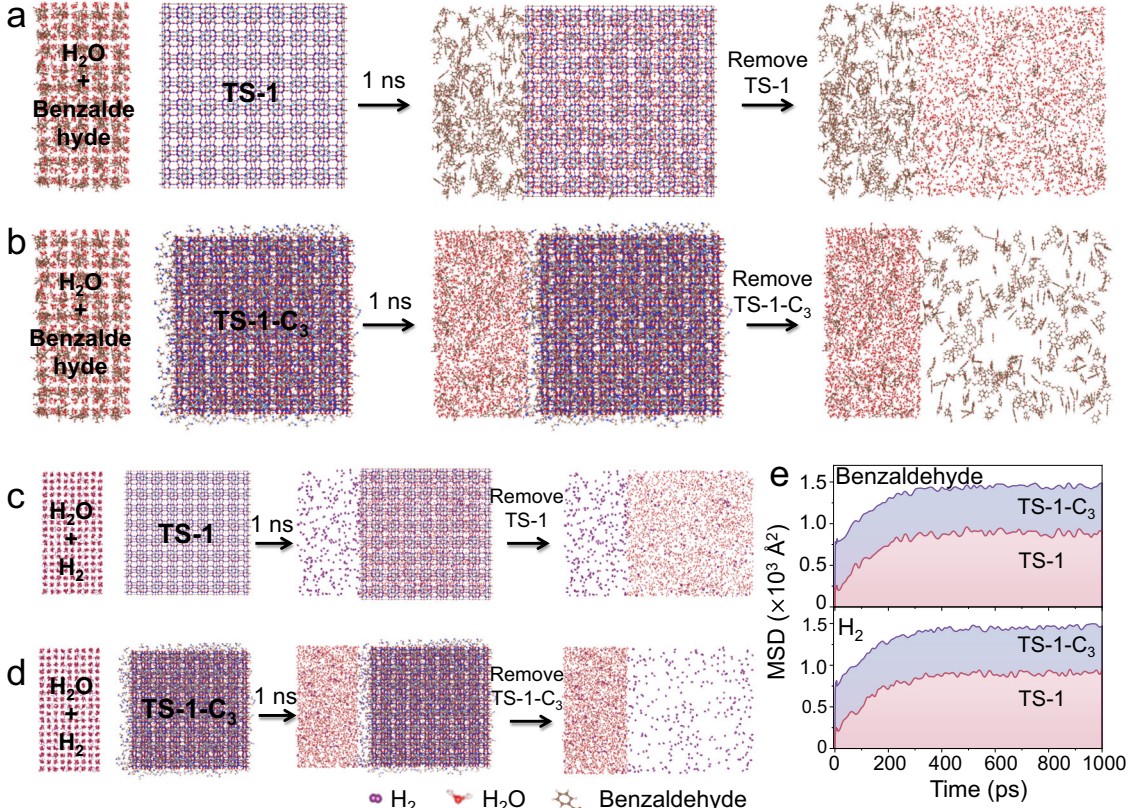

**Fig. 7 | MD simulation.** The snapshot of the penetration of benzaldehyde into (**a**) TS-1 and (**b**) TS-1-C$_3$ in aqueous solution. The snapshot of the penetration of H$_2$ into (**c**) TS-1 and (**d**) TS-1-C$_3$ in aqueous solution. **e** Mean-squared displacements (MSD) of benzaldehyde (top) and H$_2$ (bottom) in TS-1 and TS-1-C$_3$ systems with water as a solvent.

diffractometer using a copper target (Cu Kα). N$_2$ physisorption isotherms were performed on a Micromeritics Tristar II 3020 instrument at −190 °C. Surface areas, micropore volumes and pore size distributions were analyzed by Brunauer-Emmett-Teller (BET) method, t-plot analysis and Barrett-Joyner-Halenda (BJH) method, respectively. The spectra were processed and analyzed by the software codes Athena and Artemis. Fourier transform infrared (FT-IR) spectroscopic analysis was conducted by using pressed KBr disk in the region of 4000–400 cm$^{-1}$. The contact angles were measured using a contact angle system (OCA 20, Dataphysics) at ambient temperature, with the probe liquid being 10 µL. For sample preparation, 50 mg of the catalyst was compressed into a wafer with a diameter of 13 mm and a smooth surface under a pressure of 10 MPa. Contact angle images were taken after the application of the liquid droplet on the surface of samples (or the application of H$_2$ gas bubble on the surface of samples underwater).

**Catalytic performance measurements**

The liquid-phase hydrogenation of aldehydes/ketones was performed in a glass round-bottom flask reactor. The substrate was dispersed in water and uniformly distributed by ultrasound, followed by the addition of catalyst. The air in reactor is removed by vacuuming and then H$_2$ (1 atm) was injected. The reaction was initiated by heating to the designated temperature under stirring at 900 rpm. After the reactions, the organic phase was extracted from the aqueous phase using ethyl acetate, with an extraction-to-solvent ratio of 2:1 (v/v). The conversion and yield were analyzed by a gas chromatography spectrometry (GC-2010 Plus, MXT-1 column). External standard method was used for the product quantification in the current study. The carbon balance of all examined catalysts was in the range of 96 to 99%. In the D$_2$O-labeling reaction experiments, the reaction procedure is the same as that of benzaldehyde hydrogenation, except that D$_2$O is used instead of H$_2$O.

The catalytic reaction data were calculated based on the following formulas:

$$\text{Conversion}(\%) = \left( 1 - \frac{\text{molar amount of substrate after reaction}}{\text{initial molar amount of substrate fed}} \right) \times 100\% \quad (1)$$

$$\text{Yield}(\%) = \frac{\text{molar amount of one product}}{\text{initial molar amount of substrate fed}} \times 100\% \quad (2)$$

**Reporting summary**

Further information on research design is available in the Nature Portfolio Reporting Summary linked to this article.

## Data availability

All relevant data that support the findings of this study are presented in the manuscript and supplementary information file. Source data are provided with this paper.

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

## Acknowledgements

We acknowledge financial support from the Shandong Excellent Young Scientists Fund Program (Overseas, 2022HWYQ-082), the National Natural Science Foundation of China (Grant No. 22278174, 21808079), Jinan Science and Technology Bureau (2021GXRC086) and the financial support from the Shandong Provincial Natural Science Foundation (Grant No. ZR2023MB103, ZR202102230042).

## Author contributions

S.W. conceived and designed experiment under the supervision of D.G. and G.C. D.G. and G.C. supervised and led the project. S.W., J.R. and Y.L. supported for catalyst synthesis and testing. R.H. and X.J. performed DFT calculation. L.S. supported for analyzing the X-ray absorption spectroscopy. H.Z. supported for conducting and analyzing the MS spectra in $D_2O$-labeling experiments. S.W., J.R., Y.L., D.G. and G.C. wrote the manuscript. All authors discussed the results, contributed towards data interpretation and commented on the manuscript.

## Competing interests

The authors declare no competing interests.
