## [Peer Review File · Nature Communications]

Surface hydrophobization of zeolite enables mass transfer matching in gas-liquid-solid three-phase hydrogenation under ambient pressureREVIEWER COMMENTS

Reviewer #1 (Remarks to the Author):

In this article, the authors report a gas-liquid-solid three-phase hydrogenation catalyst with enhanced mass transfer by controlling the wettability and pore microenvironment of the zeolite catalyst. The effects of hydrophobic/aerophilic property on the mass transfer of H₂ and substrate molecules have been investigated. This is a very interesting topic. I believe that upon modifications this manuscript can be suitable for publication in Nature Communications. The additional comments are as follows.

1. It is not straight forward at all labels such as -C3, -C6 and -C16 groups of Pt@TS-1-Cx. They are only described in the supplementary information. The main manuscript should briefly describe the meaning of the detailed catalysts, especially the surface modified groups.
2. On page 9, in the XAS characterization section, it states that "the Pt species in TS-1 zeolites primarily exist as single Pt atoms bonded to the O atoms, along with a minor quantity of Pt NPs". What is the main activation mode of H₂ molecules, homolytic cleavage or heterolytic cleavage during the hydrogenation process of aldehydes/ketones? The authors should discuss this point in more detail in catalytic results section, because the H₂ activation mode may be changed after modifying the external zeolite surface with organosilanes.
3. In the D₂O-labeling reaction experiments (Fig. 5), the author used D₂O instead of H₂O, but deuterated products appeared in the hydrogenation products. Why do deuterium atoms appear in the products of hydrogenation reactions with H₂ as the hydrogen source?
4. Is it possible for -C3, -C6 and -C16 groups to enter the micropores of zeolite, and further influence the diffusion of the H₂ and reactants in the microchannel due to the different dimension of these groups?
5. After modifying with silanes on the zeolite surface, the H₂ and benzaldehyde molecules diffuse more easily through the pore apertures of TS-1-C3 than TS-1 obtained from molecular dynamic simulation. Detailed diffusion rate of these molecules, or other parameters to quantitatively reflect the diffuse property should be provided.
6. Increasing gas pressure is a simple way to increase H₂ solubility in solvents. What will be the catalytic activity of Pt@TS-1-C3 and Pt@TS-1 catalysts when increasing pressure? In that case, does Pt@TS-1-C3 still have the same advantage?

Reviewer #2 (Remarks to the Author):

The authors report on the aldehydes/ketones hydrogenation on a zeolite nanoreactor with joint gas-solid-liquid interfaces and controlled wettability. This catalytic nanoreactor with aerophilic/hydrophobic property can promote the diffusion of H₂ and the mass transfer of reactant/product molecules. Through this, the hydrogenation of aldehydes/ketones can be operated in water under ambient H₂ pressure, and result in a noteworthy turnover frequency. The reasons of excellent catalytic performance were revealed by experiment and theoretical calculation. This is a significant investigation and the experiments have been conducted adequately, so I recommend this manuscript for publication after addressing the following comments.

1. I am curious about the wettability comparison of Pt@TS-1-C3 and Pt@TS-1 by confocal laser scanning microscopy (CLSM), please explain this in more detail, especially to specify the type of fluorescent agent used.
2. In the discussion of the reaction mechanism, under high H₂ pressure, the driving force of H₂ molecules entering the catalyst channels can be the pressure difference between the inside and outside channel. However, this work highlights the catalytic conversion under ambient pressure, and it is necessary for the authors to explain what the driving force is for H₂ to enter zeolite pores smoothly.
3. In this work, the zeolite was modified by the -C3, -C6 and -C16 groups. Can these groups with large chain length affect the diffusion of the reactant molecules in the micro/mesopore channels during the reaction?
4. There are missing experimental details. The catalyst characterization procedures should be described in more detail, such as the D₂O-labeling reaction experiments.
5. In the Cs-corrected HAADF STEM images (Fig. 1d), the authors use red circles on the images to

highlight Pt nanoparticles. However, it is difficult to see the presence of obvious Pt nanoparticles.

Reviewer #3 (Remarks to the Author):

In this manuscript, the authors report a nanoreactor with joint gas-solid-liquid interfaces and controlled wettability. This design facilitates the liquid-phase hydrogenation of aldehydes or ketones into their respective alcohols. Through the introduction of organosilanes, the zeolite surface is rendered hydrophobic and aerophilic. This modification enhances substrate retention on the zeolite and facilitates improved hydrogen gas penetration, leading to a 4.3-fold enhancement in the benzaldehyde hydrogenation efficiency.

In my opinion, certain experimental results presented in this paper do not convincingly support the stated experimental conclusions. After carefully reviewing the manuscript, I came to the conclusion that this draft is not appropriate for publication in Nature communications due to the following concerns.

1. The synthesis method employed in this article lacks novelty. Several previous studies, such as "Hydrothermal synthesis and catalytic performance of bulky titanium silicalite-1 aggregates assembled by bridged organosilane" and "One-step synthesis of hybrid zeolite with exceptional hydrophobicity to accelerate the interfacial reaction at low temperature" have utilized nearly the same approach to modify the hydrophilic and hydrophobic properties of zeolites.

2. In Fig. 1d, the authors have circled some regions indicating that Pt single atoms exist in those regions. However, in my view, most of the regions are not clear enough to confirm the existence of Pt single atoms. Additionally, there are many circled areas that are not different from other uncircled areas. The authors should provide clearer images to substantiate their conclusion.

3. As noted by the authors, Fig. 1e and Fig. 1f are high-magnification Cs-corrected HAADF STEM images of Pt@TS-1 zeolite. However, it seems that Fig. 1f is the magnified image of Fig. 1e in the same sample. But according to the scale in the figure, the sizes of the zeolite are different. The accuracy of the images is questionable. Additionally, in Fig. 1f, the authors have marked the positions of the Pt single atoms, but these may be just parts of the zeolite. To substantiate this claim, the authors should provide compelling evidence, such as a comparison between zeolites with and without Pt single atoms.

4. The statement "a 4.3-fold increase in reaction rate compared to unmodified catalyst" is based on a comparison with samples synthesized by the authors themselves. To demonstrate the superior performance of their catalyst in hydrogenation reactions, the authors should provide a comparative analysis with other catalysts used in this field and present this information in a table or chart for clarity and context.

5. In the first paragraph of the results and discussions section, the sentence "Then, rendering the external surface of the zeolite hydrophobic by appending organosilanes, namely Pt@TS-1-Cx (Cx represents the organic substituent of the silane sheath)." appears to be incomplete. The authors should check the manuscript carefully.

Reviewer #4 (Remarks to the Author):

The authors reported a zeolite nanoreactor with joint gas-liquid-solid interface. The micro environment of the catalytic site was modified by surface hydrophobization to enhance the concentration of H₂ and aldehydes/ketones substrate. Some interesting results and mechanism understanding are obtained. However, the concept of enhancing the performances of heterogeneous catalysts by constructing hydrophobic or hydrophilic surface has been reported in several papers, for example in ref. 12-17 as mentioned by the authors. In particular, very similar ideas were reported in refs. 15 and 16 (Mi et al., JACS, 2017, 139, 10441; Li et al., AM Interfaces, 2018, 5, 1801259). This decreases the novelty of the present work. It is more suitable to more specific journals on catalysis. In addition, the following issues should be adequately addressed.

(1) The critical issue is the reaction evaluation and the quantitative analysis.

The authors only described that "liquid products were analyzed by a gas chromatography spectrometry", how did the author calculate the conversion? Is it based on the relative peak area of benzaldehyde and benzyl alcohol? Did the authors use internal standard chemical and calculate

the carbon balance?

It should be noted that the water solubility of benzaldehyde (BA) is extremely low and the concentration of BA (1%) used in the current work would lead to the formation of oil-water phase, which could greatly affect the accuracy of quantitative analysis.

The catalyst weight for each reaction was about 58 mg, and only 50 mg BA reactant was added for each reaction. It should be noted that the BA reactant (lipophilic) can be adsorbed on the catalyst especially on the lipophilic Pt@TS-1-Cx, which was also investigated by the authors in Supplementary Figure 13. It is possible that the BA in the oil-water phase is adsorbed on the catalyst but not completely dissolved in the water phase (only BA dissolved in water phase can be detected by GC). The above issue could lead to the overestimate of BA conversion.

Experimental details for the reaction evaluation and the quantitative analysis should be fully addressed. Carbon balance for each reaction should be carefully calculated. To enhance the reliability of the data, error bars (based on several repeated experiment) for some important reaction data is suggested to be added.

(2) The details of the contact angle measurement should be provided. Is the catalyst in Fig. 3b,e compressed into slice? If so, how the authors assure it has similar wetting properties with the particulate catalysts. In addition, scale bars should be added in Fig. 3b,e and related supplementary figures.

(3) The authors only investigate the catalyst at 50 °C. The surface wetting property may be sensitive to the temperature. In what temperature range does the catalyst maintain this high performance?

(4) The authors carried MD without considering the effect water molecules, which is different from the real situation. This gap should be addressed.

(5) How did the author calculate the adsorption capacity (Supplementary Figure 13)? How was this adsorption experiment carried out? Experimental details must be addressed.

RESPONSE TO REVIEWERS' COMMENTS

Response to Reviewer #1

In this article, the authors report a gas-liquid-solid three-phase hydrogenation catalyst with enhanced mass transfer by controlling the wettability and pore microenvironment of the zeolite catalyst. The effects of hydrophobic/aerophilic property on the mass transfer of H₂ and substrate molecules have been investigated. This is a very interesting topic. I believe that upon modifications this manuscript can be suitable for publication in *Nature Communications*. The additional comments are as follows.

Response: We thank the reviewer very much for his/her positive comments. These thoughtful comments definitely help us to further improve the quality of our manuscript. After carefully considering the reviewer #1's comments, we have implemented the following revisions: (1) Description of the meaning of the catalyst labels in the manuscript. (2) Understanding of the H₂ activation mode in combination with DFT calculations. (3) Explanation of the mechanism of D₂O-labeling reaction experiments. (4) Supplement of the reason why organic silane does not affect the diffusion of molecules within zeolite micropores. (5) Quantitative comparison of the differences in diffusion rates in the MD simulation. (6) Investigation of the influence of H₂ pressure on catalytic performance.

Comment 1. It is not straight forward at all labels such as -C₃, -C₆ and -C₁₆ groups of

Pt@HieTS-1-C_x. They are only described in the supplementary information. The main manuscript should briefly describe the meaning of the detailed catalysts, especially the surface modified groups.

Response: Thank you for this suggestion. We have elaborated on the meaning of catalyst labels in the revised manuscript.

Actions taken: The revisions made are as follows:

“Then, the external-surface hydrophobization of zeolite is obtained by modification with organosilanes. The catalysts are denoted as Pt@HieTS-1-C_x, where the @ symbol is related to the encapsulation of the Pt active sites within the hierarchical HieTS-1 zeolite and C_x represents the organic substituent of the silane. Specifically, trimethoxy(propyl)silane, trimethoxyphenylsilane and hexadecyltrimethoxysilane are labeled as -C₃, -C₆, and -C₁₆, respectively.” (Line 14-20, Page 5, in revised manuscript)

Comment 2. On page 9, in the XAS characterization section, it states that “the Pt species in HieTS-1 zeolites primarily exist as single Pt atoms bonded to the O atoms, along with a minor quantity of Pt NPs”. What is the main activation mode of H₂ molecules, homolytic cleavage or heterolytic cleavage during the hydrogenation process of aldehydes/ketones? The authors should discuss this point in more detail in catalytic results section, because the H₂ activation mode may be changed after modifying the external zeolite surface with organosilanes.

Response: Thanks for your very specialized question.

Since the Pt NPs and atomically dispersed Pt species both exist in the as-

synthesized catalyst, the H₂ activation can be conducted via both homolysis and heterolysis models. As for Pt NPs, dihydrogen molecules first adsorb on metallic Pt NPs, followed by barrierless homolytic dissociation into the neutral β -hydride species Pt-H.^{1, 2} With regard to atomically dispersed Pt ^{δ +} species, the type of dihydrogen heterolysis is the main activation model³⁻⁶, where the dihydrogen can be dissociated by Pt ^{δ +}...O²⁻ pairs. The activated atomic hydrogen species (H ^{δ -} and H⁺) will be further utilized in hydrogenations.

In addition, we also employed density functional theory (DFT) calculations to comprehend the energy barrier for dihydrogen activation on Pt NPs and atomically dispersed Pt ^{δ +} species (Fig. R1). Consistent with the above discussion, the dissociation H₂ follows a heterolytic pathway on atomically dispersed Pt ^{δ +}, while it manifests homolytic dissociation on the metallic Pt (111) surface. Furthermore, the barrier of H₂ dissociation on atomically dispersed Pt ^{δ +} species (0.71 eV) is higher than that on metallic Pt (111) surface (0.43 eV) (Fig. R1). This implies that the primary pathway for dihydrogen activation is heterolysis dissociation on Pt NPs, while the heterolytic dissociation on atomically dispersed Pt ^{δ +} acts as an assistant role.

After modifying the external zeolite surface with organosilanes, this modification does not influence the mode of dihydrogen activation. Because the organic silanes exclusively modify the external surface of zeolites, while the Pt active sites are encapsulated within the zeolite, thus the activation of H₂ predominantly is carried out within the microporous channels of zeolites.

Fig. R1 Relative energy plots of H₂ dissociation on metallic Pt (111) plane and atomically dispersed Pt species.

Actions taken: Fig. R1 has been moved to Supplementary Fig. 22 of the revised Supplementary Information. Corresponding explanation has been added in the revised Supplementary Information. (Page 30, in revised Supplementary Information)

Reference

- [1] Chai, Y. et al. Acetylene-selective hydrogenation catalyzed by cationic nickel confined in zeolite. *J. Am. Chem. Soc.* **141**, 9920–9927 (2019).
- [2] Kuai, L. et al. Titania supported synergistic palladium single atoms and nanoparticles for room temperature ketone and aldehydes hydrogenation. *Nat. Commun.* **11**, 48 (2020).
- [3] Whittaker, T. et al. H₂ oxidation over supported Au nanoparticle catalysts: evidence for heterolytic H₂ activation at the metal–support interface. *J. Am. Chem. Soc.* **140**, 16469–16487 (2018).
- [4] Deng X. et al. Zeolite-encaged isolated platinum ions enable heterolytic dihydrogen activation and selective hydrogenations. *J. Am. Chem. Soc.* **143**,

20898-20906 (2021).

- [5] Huang, Z. Q. et al. Understanding all-solid frustrated-Lewis-pair sites on CeO₂ from theoretical perspectives. *ACS Catal.* **8**, 546–554 (2018).
- [6] Riley, C. et al. Design of effective catalysts for selective alkyne hydrogenation by doping of ceria with a single-atom promotor. *J. Am. Chem. Soc.* **140**, 12964–12973 (2018).

Comment 3. In the D₂O-labeling reaction experiments (Fig. 5), the author used D₂O instead of H₂O, but deuterated products appeared in the hydrogenation products. Why do deuterium atoms appear in the products of hydrogenation reactions with H₂ as the hydrogen source?

Response: Thanks for your comments. After using D₂O instead of H₂O for benzaldehyde hydrogenation, the product of benzyl alcohol was obviously deuterated, which can be attributed to the H-D exchange with the assistance of water (Fig. R2)¹⁻³. Fig. 5b presented the signals of deuterated molecules in the MS spectra for benzyl alcohol product catalyzed by Pt@HieTS-1. This provides evidence that the D₂O-involved D-H exchange could serve as a possible pathway for the hydrogenation of aldehydes/ketones.

Fig. R2 The schematic diagram of two possible reaction pathways for the hydrogenation of aldehydes/ketones over Pt@HieTS-1.

Reference

- [1]. Ananyev, M. V., Farlenkov, A. S., & Kurumchin, E. K. Isotopic exchange between hydrogen from the gas phase and protonconducting oxides: theory and experiment. *Int. J. Hydrogen Energy* **43**, 13373–13382 (2018).
- [2]. Dai, Y. et al. Water-enhanced selective hydrogenation of cinnamaldehyde to cinnamyl alcohol on RuSnB/CeO₂ catalysts. *Appl. Catal., A* **582**, 117098 (2019).
- [3]. Farlenkov, A. S., Zhuravlev, N. A., Denisova, T. A., & Ananyev, M. V. Interaction of O₂, H₂O and H₂ with proton-conducting oxides based on lanthanum scandates. *Int. J. Hydrogen Energy* **44**, 26419–26427 (2019).

Comment 4. Is it possible for -C₃, -C₆ and -C₁₆ groups to enter the micropores of zeolite, and further influence the diffusion of the H₂ and reactants in the microchannel due to the different dimension of these groups?

Response: We thank the reviewer for his/her insightful comment. During the modification process of zeolite, the precursor of organosilane is incapable of entering zeolite micropores, because the micropore diameter of HieTS-1 zeolite is so small that

organic silane molecules with larger molecular dynamic diameters are unable to enter these micropores (Fig. R3 and Fig. R4).¹⁻³ Consequently, organic silanes such as -C₃, -C₆, and -C₁₆ groups can only be utilized to modify the external surface of the zeolite. Therefore, this modification does not affect the diffusion of H₂ or substrate molecules within the micropores of the zeolite. Nevertheless, it does have an impact on the diffusion of H₂ or substrate molecules from the outside of zeolite into the channels of zeolite. It can be supported by evidences from in situ DRIFTS spectra of benzaldehyde adsorbed (Fig. 3d) and adsorption experiments (Supplementary Fig. 21).

Fig. R3 The three-dimensional molecular dimensions of the (a) trimethoxy(propyl)silane (-C₃), (b) trimethoxyphenylsilane (-C₆) and (c) hexadecyltrimethoxysilane (-C₁₆).

Fig. R4 The channels of MFI framework type zeolite (Å). Data from the Database of Zeolite Structures (<http://www.iza-structure.org/databases/>).

Reference

- [1] Olson, D. H., Kokotailo, G. T., Lawton, S. L., & Meier, W. M. Crystal structure and structure-related properties of ZSM-5. *J. Phys. Chem.* **85**, 2238–2243 (1981).
- [2] Breck, D. W. Zeolite molecular sieves. *Wiley: New York* (1974).
- [3] Burggraaf, A. J., Vroon, Z. A. E. P., Keizer, K., & Verweij, H. Permeation of single gases in thin zeolite MFI membranes. *J. Membr. Sci.* **144**, 77–86 (1998).

Comment 5. After modifying with silanes on the zeolite surface, the H₂ and benzaldehyde molecules diffuse more easily through the pore apertures of TS-1-C₃ than TS-1 obtained from molecular dynamic simulation. Detailed diffusion rate of these molecules, or other parameters to quantitatively reflect the diffuse property should be provided.

Response: Thanks for your excellent suggestions. Based on the molecular dynamic (MD) simulation, the penetration behavior of the benzaldehyde and H₂ molecules across the TS-1 and TS-1-C₃ frameworks was quantitatively compared by employing penetration ratio as a descriptor. Specifically, the penetration ratio denotes the ratio of the number of molecules diffusing into the zeolite channels to the initial number of molecules in the simulation. As depicted in Table R1, the penetration ratios of benzaldehyde and H₂ molecules in the TS-1-C₃ framework are approximately threefold higher than those in TS-1.

Table R1. Simulation data of the penetration ratio of benzaldehyde/H₂ molecules for TS-1 and TS-1-C₃.

Entry	Molecule	Zeolite	Penetration ratio ^a
1		TS-1	24.2%
2		TS-1-C ₃	72.3%
3		TS-1	23.3%
4	H ₂	TS-1-C ₃	75.8%

^a Penetration ratio is defined as the ratio of the number of benzaldehyde (or H₂) molecules diffusing into the zeolite channels to the initial number of benzaldehyde (or H₂) molecules.

Actions taken: Table R1 has been moved to Supplementary Table 6 in the Supplementary Information. And the corresponding explanation was made in the revised manuscript, as follow:

“As depicted in Supplementary Table 5, the penetration ratios of benzaldehyde and H₂ molecules in the TS-1-C₃ framework are approximately threefold higher than those in TS-1. This comparison further substantiates that benzaldehyde and H₂ molecules manifest a heightened propensity for diffusion into the zeolite channels within the TS-1-C₃ framework.” (Line 14-18, Page 14, in revised manuscript)

Comment 6. Increasing gas pressure is a simple way to increase H₂ solubility in solvents. What will be the catalytic activity of Pt@HieTS-1-C₃ and Pt@HieTS-1 catalysts when increasing pressure? In that case, does Pt@HieTS-1-C₃ still have the same advantage?

Response: Thanks for your comments. We agree with the viewpoint of reviewer #1 that

increasing hydrogen pressure is a straightforward method to enhance H₂ solubility. Taking Pt@HieTS-1 catalyst as an example, increasing the H₂ pressure from 1 atm to 1 MPa can enhance the reaction rate from 0.16 h⁻¹ to 0.54 h⁻¹ (Fig. 4f). Furthermore, the catalytic assessments of Pt@HieTS-1-C₃ and Pt@HieTS-1 were also carried out under a broader range of H₂ pressures (Fig. R5). The catalytic activity of Pt@HieTS-1-C₃ and Pt@HieTS-1 were both improved after the increase of H₂ pressure, primarily attributed to the heightened solubility of H₂. In particular, under identical H₂ pressure, the catalytic activity of Pt@HieTS-1-C₃ surpasses that of Pt@HieTS-1 significantly. This can be attributed to the unique hydrophobic/lipophilic of the silane sheath on Pt@HieTS-1-C₃, which promotes the mass transfer of lipophilic substrates to the active sites and facilitates the diffusion of hydrophilic products from the catalyst into the aqueous solvent.

Fig. R5 The effect of the H₂ pressure on benzaldehyde hydrogenation. Reaction condition: benzaldehyde (0.47 mmol), Pt dosage (2.1×10^{-3} mmol), water (5 mL), 50 °C, 0.5 h.

Actions taken: Fig. R5 has been moved to Supplementary Fig. 15 in the revised

Supplementary Information. The revisions made in the revised manuscript are as follows:

“To further validate this notion, another reaction was conducted with Pt@HieTS-1 as the catalyst while raising the H₂ pressure to 1 MPa (Fig. 4f). The result revealed that Pt@HieTS-1 exhibited a comparable reaction rate under elevated H₂ pressure to that of Pt@HieTS-1-C₃ under ambient pressure. Furthermore, the catalytic activities of Pt@HieTS-1-C₃ and Pt@HieTS-1 were both improved after the increase of H₂ pressure, primarily attributed to the heightened solubility of H₂ (Supplementary Fig. 15). In particular, under identical H₂ pressure, the catalytic activity of Pt@HieTS-1-C₃ surpasses that of Pt@HieTS-1 significantly. This can be attributed to the unique hydrophobicity of the silane sheath in Pt@HieTS-1-C₃, which promotes the mass transfer of hydrophobic substrates to the active sites and facilitates the diffusion of hydrophilic products from the catalyst into the aqueous solvent.” (Line 10-16, Page 12, in revised manuscript)

Response to Reviewer #2

The authors report on the aldehydes/ketones hydrogenation on a zeolite nanoreactor with joint gas-solid-liquid interfaces and controlled wettability. This catalytic nanoreactor with aerophilic/hydrophobic property can promote the diffusion of H₂ and the mass transfer of reactant/product molecules. Through this, the hydrogenation of aldehydes/ketones can be operated in water under ambient H₂ pressure, and result in a noteworthy turnover frequency. The reasons of excellent catalytic performance were revealed by experiment and theoretical calculation. This is a significant investigation and the experiments have been conducted adequately, so I recommend this manuscript for publication after addressing the following comments.

Response: We thank the reviewer very much for reviewer #2's positive recommendation and insightful comments. The reviewer #2's comments definitely help us to improve the quality of our manuscript. After carefully considering the reviewer #2's comments, we have made extensive revision to improve the quality of our manuscript, as follow: (1) Elucidation of the wettability comparison mechanism by CLSM. (2) Discussion of the driving force for H₂ to enter zeolite pores. (3) Examination of the impact of organosilanes on substrate diffusion. (4) More experimental details were supplemented. (5) Confirmation of the existence of Pt NPs.

Comment 1. I am curious about the wettability comparison of Pt@HieTS-1-C₃ and Pt@HieTS-1 by confocal laser scanning microscopy (CLSM), please explain this in more detail, especially to specify the type of fluorescent agent used.

Response: Thanks for your suggestion. Since the water contact angle test only displays the hydrophobic property in macroscopic manner, confocal laser scanning microscopy (CLSM) studies can be used to investigate the hydrophobic property at micro-nano level. Water-soluble fluorescent dye (Fluorescein Sodium) was selected as the fluorescent probe molecule. In this regard, fluorescent dye can only lighten the area where water is present. Taken Pt@HieTS-1-C₃ and Pt@HieTS-1 as contrast examples, the schematic diagram and the CLSM images are shown in Fig. 2c. It can be seen that fluorescent dye can lighten the whole Pt@HieTS-1 body, whereas very weak fluorescence signal is observed on Pt@HieTS-1-C₃ body, indicating that water cannot penetrate into the channels of Pt@HieTS-1-C₃.

Actions taken: The revisions made in the revised manuscript are as follows:

“Water-soluble fluorescent dye (Fluorescein Sodium) was selected as the fluorescent probe molecule. In this regard, fluorescent dye can only lighten the area where water is present.” (Line 15-17, Page 8, in revised manuscript)

Comment 2. In the discussion of the reaction mechanism, under high H₂ pressure, the driving force of H₂ molecules entering the catalyst channels can be the pressure difference between the inside and outside channel. However, this work highlights the catalytic conversion under ambient pressure, and it is necessary for the authors to explain what the driving force is for H₂ to enter zeolite pores smoothly.

Response: Thanks for your comment. In this work, the silane sheath imparts hydrophobic/aerophilic properties for Pt@HieTS-1-C_x, serving as the driving force for

the unimpeded entry of H₂ into the zeolite channels and markedly increasing the H₂ concentration within zeolite channels. Such structure allows H₂ gas to be filled and stored inside its mesopores and micropores. In addition, the hydrophobic/aerophilic Pt@HieTS-1-C_x is able to float to the surface of aqueous solution during the stirring, as a result, it can be refilled H₂ from upper H₂ atmosphere when H₂ is consumed.

Comment 3. In this work, the zeolite was modified by the -C₃, -C₆ and -C₁₆ groups. Can these groups with large chain length affect the diffusion of the reactant molecules in the micro/mesopore channels during the reaction?

Response: Thanks for your comment. In this study, different silane sheaths have minimal differences in the diffusion of H₂ gas. However, their influence on substrate molecules diffusion into channels becomes more pronounced. It can be confirmed from dynamic contact angle tests, spectroscopy and adsorption characterization, involving H₂ gas-bubble contact angles, in situ DRIFTS spectra of benzaldehyde adsorption and adsorption experiment of benzaldehyde.

- (1) **The influence on the diffusion of H₂ gas.** As shown in Fig. 3e, Pt@HieTS-1-C₃, Pt@HieTS-1-C₆, and Pt@HieTS-1-C₁₆ exhibited superaerophilic properties with respect to the underwater H₂ bubbles. The bubbles displayed bursting behavior and were able to diffuse completely within merely 30 ms, indicating that H₂ gas can rapidly diffuse into the channels of the -C₃, -C₆ and -C₁₆ groups modified zeolite.
- (2) **The influence on the diffusion of substrate molecule.** In the hydrogenation performance of the organosilane-modified zeolite catalysts, the reactivity of

Pt@HieTS-1-C₃ is 4.3 times superior to that of Pt@HieTS-1 (Fig. 4). Further increasing the carbon number of the modified alkyl leads to a lower enhancement effect, most likely due to the blockage of benzaldehyde molecule diffusion caused by the long carbon-chain alkyl. To further confirm the variations in the adsorption capabilities of different silane sheaths, in situ DRIFTS spectra of benzaldehyde adsorption was employed (Fig. 3d). With the carbon chain length of organosilane precursor extends, it is slightly difficult for benzaldehyde to enter the pores of zeolite because of limited space, but it can still be adsorbed by the Pt active sites inside the zeolite. The adsorption experiments of benzaldehyde also confirmed this point, as the carbon chain length of the silane sheath increases, the catalyst's adsorption capability for benzaldehyde decreases (Fig. R6). Nevertheless, the catalyst modified with the -C₁₆ group still exhibits higher adsorption capacity compared to the unmodified catalyst.

Updated Fig. 4 (a) The evolution of benzaldehyde conversion with the reaction time on various catalysts. (b) The kinetic plots of the catalysts, calculated at the low conversion of benzaldehyde. (c) The corresponding rate constant value of various catalysts.

Fig. 3d In situ DRIFTS spectra of benzaldehyde adsorbed on Pt@HieTS-1, Pt@HieTS-1-C₃, Pt@HieTS-1-C₆ and Pt@HieTS-1-C₁₆.

Fig. R6 Adsorption capacity of benzaldehyde (or benzyl alcohol) over different catalysts.

Comment 4. There are missing experimental details. The catalyst characterization procedures should be described in more detail, such as the D₂O-labeling reaction experiments.

Response: Thanks for your reminding. The detailed experimental information of D₂O-

labeling reaction experiments has been added in the revised manuscript with the words marked in blue. (Line 18-21, Page 18, in revised manuscript)

Comment 5. In the Cs-corrected HAADF STEM images (Fig. 1d), the authors use red circles on the images to highlight Pt nanoparticles. However, it is difficult to see the presence of obvious Pt nanoparticles.

Response: Thanks for your comment. The presence of Pt nanoparticles (NPs) can be confirmed by the TEM images. As shown in Fig. 1b and Fig. R7, it is obvious that a large number of Pt NPs are distributed inside the zeolite.

Fig. 1b HRTEM image of Pt@HieTS-1 zeolite.

Fig. R7 HRTEM image of Pt@HieTS-1 zeolite.

Actions taken: Fig. R7 has been added to Supplementary Fig. 2 of the revised Supplementary Information. In addition, we have replaced Fig. 1d with high-quality images in order to provide a clearer view of Pt NPs.

Response to Reviewer #3

In this manuscript, the authors report a nanoreactor with joint gas-solid-liquid interfaces and controlled wettability. This design facilitates the liquid-phase hydrogenation of aldehydes or ketones into their respective alcohols. Through the introduction of organosilanes, the zeolite surface is rendered hydrophobic and aerophilic. This modification enhances substrate retention on the zeolite and facilitates improved hydrogen gas penetration, leading to a 4.3-fold enhancement in the benzaldehyde hydrogenation efficiency.

In my opinion, certain experimental results presented in this paper do not convincingly support the stated experimental conclusions. After carefully reviewing the manuscript, I came to the conclusion that this draft is not appropriate for publication in *Nature communications* due to the following concerns.

Response: We appreciate the reviewer for careful and thoughtful comments, which are valuable in improving the quality of our manuscript. We have carefully addressed each issue raised in review. As described below, we have made a detailed explanation and performed additional evidences to address the reviewer's concerns: (1) The different methods of catalyst preparation between this work and reported literatures were compared, and the innovation points of this work was expounded. (2) The existence and precise location of Pt single atoms have been substantiated by DFT calculation and additional evidence. (3) The mistake of scale bar in the Cs-corrected iDPC STEM image was corrected. (4) The catalytic properties of reported catalysts in this field were summarized in a table. (5) Grammatical mistakes in the manuscript were corrected.

These revises address the concerns raised by the reviewer and fortify the overall clarity and reliability of our work.

Comment 1. The synthesis method employed in this article lacks novelty. Several previous studies, such as “**References 1:** *Chinese Journal of Catalysis* 39 (2018) 275–282” and “**References 2:** *Microporous and Mesoporous Materials* 280 (2019) 195–202” have utilized nearly the same approach to modify the hydrophilic and hydrophobic properties of zeolites.

Response: Thanks for your insightful comments.

According to the references mentioned by the reviewers, in *References 1*, a bridged organosilane (1,8-bis(triethoxysilyl)octane) was used as a silanization agent to assemble TS-1 nanoseeds into large aggregates (Fig. R8).¹ The main function of organosilane in this reference was to prevent the aggregation of zeolite crystals during crystallization. The crystallization of silanized-seeds and nanocrystallites led to large and irregular TS-1 zeolite aggregates, based on the special sol-gel chemistry of bridged organosilane. In *References 2*, the author synthesized organic-inorganic hybrid zeolite by one-step synthesis using dimethyldiethoxysilane as organic Si source (Fig. R9).² Both organosilicon and inorganic silicon precursors were added to the initial gel before the crystallization and growth of zeolite.

In contrast to the above references, we employed a post-silylation method to modify organosilane onto the zeolite surface (Fig. R10). Furthermore, the innovation of our manuscript is not only focus on the synthesis of Pt@HieTS-1-C_x with surface

hydrophobization, but mainly on the mass transfer matching in gas-liquid-solid three-phase hydrogenation under ambient pressure. As we all know, molecular mass transfer in gas-liquid-solid three-phase hydrogenation reactions encompasses the transfer of gas molecules, substrate molecules, and product molecules. However, the simultaneous regulation of the diffusion of these molecules to achieve mass transfer matching in gas-liquid-solid three-phase hydrogenation reactions is currently unexplored.

In this work, the Pt active sites are in-situ encapsulated within the HieTS-1 zeolite, followed by modifying the external zeolite surface with organosilanes. The silane sheath with aerophilic/hydrophobic property can promote the diffusion of H₂ and the mass transfer of reactant/product molecules. Specifically, in aqueous solutions, the gaseous H₂ molecules can rapidly diffuse into the zeolite channels, thereby augmenting concentration surround Pt sites. Simultaneously, the Pt@HieTS-1-C_x, endowed with lipophilic attributes, facilitates the enrichment of the substrate molecules on the catalyst, while expediting the hydrophilia alcohol products rediffusion back to the aqueous phase, thereby accelerating the gas-liquid-solid mass transfer.

Fig. R8 Schematic diagram of the bulky titanium silicalite-1 synthesis procedure.¹

Fig. R9 Schematic diagram of one-step synthesis of hybrid zeolite, according to the literature described.²

Fig. R10 Schematic diagram of post-silylation synthesis of hybrid zeolite in this work.

Reference

- [1] Chen, L. et al. Hydrothermal synthesis and catalytic performance of bulky titanium silicalite-1 aggregates assembled by bridged organosilane. *Chin. J. Catal.* **39**, 275–282 (2018).
- [2] Li, D. et al. One-step synthesis of hybrid zeolite with exceptional hydrophobicity to accelerate the interfacial reaction at low temperature. *Microporous Mesoporous Mater.* **280**, 195–202 (2019).

Comment 2. In Fig.1d, the authors have circled some regions indicating that Pt single atoms are existed in those regions. However, in my view, most of the regions are not clear enough to confirm the existence of Pt single atoms. Additionally, there are many circled areas that are not different from other uncircled areas. The authors should

provide clearer images to substantiate their conclusion.

Response: Thanks a lot for your careful review. (1) In the revision, we have removed the ambiguous circle markings and re-marked the positions of atomically dispersed Pt in Pt@HieTS-1 zeolite (Fig. 1d). (2) The zeolites with and without Pt species were also compared by high-magnification Cs-corrected HAADF STEM and Cs-corrected iDPC STEM to confirm the presence of Pt single atoms. (3) The precise location of Pt single atoms in zeolite framework was further substantiated by combining the Cs-corrected STEM images and density functional theory (DFT) calculations. The more detailed explanation please see Response to Comment 3.

Updated Fig. 1d Cs-corrected HAADF STEM image of Pt@HieTS-1 zeolite.

Actions taken: We have updated Fig. 1d in revised manuscript with high-quality image for a clearer view of atomically dispersed Pt species.

Comment 3. As noted by the authors, Fig. 1e and Fig. 1f are high-magnification Cs-corrected HAADF STEM images of Pt@HieTS-1 zeolite. However, it seems that Fig.

1f is the magnified image of Fig. 1e in the same sample. But according to the scale in the figure, the sizes of the zeolite are different. The accuracy of the images is questionable. Additionally, in Fig. 1f, the authors have marked the positions of the Pt single atoms, but these may be just parts of the zeolite. To substantiate this claim, the authors should provide compelling evidence, such as a comparison between zeolites with and without Pt single atoms.

Response: Thanks for carefully checking our manuscript and making useful comments.

(1) **About the scale bar in Fig. 1e and f.** We sincerely sorry for the error in the scale bar of the STEM images. After careful examination of the Cs-corrected STEM images, we confirm that the accurate scale bar for Fig.1e should be 2 nm rather than 1 nm (Please see Fig. 1e and f).

Fig. 1e and f High-magnification Cs-corrected iDPC STEM images of Pt@HieTS-1 zeolite.

(2) **Confirming the position of the Pt single atoms.**

In order to confirm the position of the Pt single atoms, the Cs-corrected STEM was employed. After sufficient communication with the engineers, we have verified that the Cs-corrected STEM of Pt@HieTS-1 was carried out in two modes, namely

HAADF STEM and iDPC STEM. Because during testing, the Pt@HieTS-1 zeolite has poor tolerance to high-pressure electron beams, making it difficult to obtain clear imaging of Pt species in high-magnification HAADF STEM mode. Therefore, the iDPC STEM model was used to provide a clear view of Pt species at HieTS-1 framework. Specifically, the Fig. 1d is Cs-corrected HAADF STEM image of Pt@HieTS-1 and the Fig. 1e is Cs-corrected iDPC STEM image of Pt@HieTS-1.

In the Cs-corrected HAADF STEM image of Pt@HieTS-1 (Fig. 1d), the atomically dispersed Pt species could be inferred as bright dots (based on the difference in atomic contrast: i.e., Si 28 g/mol, Ti 48 g/mol, Pt 195 g/mol).^{1,2} In order to confirm the position of Pt single atom in zeolite framework, we employed high-magnification Cs-corrected iDPC STEM. Different from HAADF STEM, iDPC STEM allows the simultaneous imaging of heavy and light elements (Fig. R12 c, d).³ In high-magnification Cs-corrected iDPC STEM images, we found the presence of guest molecules in 10-MR and white bright dots in 5- and 6-MR on the MFI framework. To verify whether the guest molecule in 10-MR channels is atomically dispersed Pt species, we also investigated Cs-corrected STEM images of sample without Pt. The samples with and without Pt both showed the signals of guest molecules (as indicated by the arrow in Fig. R11). The guest molecules can be directly visualized by iDPC STEM, but difficult by HAADF STEM (Fig. R12 e-g).^{3,4} These guest molecules may be volatile organic compounds adsorbed by HieTS-1 from the surrounding air, or adsorbed solvent molecules during the STEM sample preparation.⁵⁻⁷ Thus, the white bright dots located at the 5- and 6-MR can be distinguished as the Pt atoms, which is obviously different

from other framework atoms in the HieTS-1 zeolite. For comparison, the enlarged images of three typical local structures are provided in Fig. 1e.

Additionally, we also use the density functional theory (DFT) calculations to identify the configuration of atomically dispersed Pt species in HieTS-1 (Fig. R13). A more negative energy indicates greater stability and higher probability of existence. Comparing the binding energy of Pt atoms at T₁-T₈ sites, the energies at T₁-T₄ sites are relatively lower, indicating that Pt is more easily stabilized by these sites. Combined with the analysis of STEM, EXAFS and DFT calculation, the configuration of atomically dispersed Pt species in HieTS-1 was confirmed, where the isolated Pt^{δ+} species were triply/quadruply coordinated with the surrounding O atoms in the 5- and 6-MR of MFI framework.

Updated Fig. 1 (d) Cs-corrected HAADF STEM image of Pt@HieTS-1 zeolite. (e) High-magnification Cs-corrected iDPC STEM images of Pt@HieTS-1 zeolite. Zoomed-in areas of 1, 2, and 3 in (e) are the location of atomically dispersed Pt species in HieTS-1 zeolite framework and corresponding binding energy.

Fig. R11 (a) Cs-corrected HAADF STEM image of HieTS-1 zeolite. (b) Cs-corrected iDPC STEM image of HieTS-1 zeolite.

Fig. R12 (a) C_s -corrected HAADF STEM images of Rh@ZSM-5-H.¹ (b) C_s -corrected HAADF STEM images of Pt@Y.² (c) C_s -corrected iDPC STEM images of the adsorption of pyridines in ZSM-5.³ (d) C_s -corrected iDPC STEM images of the freshly prepared and air-exposed silicalite-1.⁴ (e) C_s -corrected iDPC STEM images of Pt species in UTL zeolite.⁵ (f) C_s -corrected iDPC STEM images of Ir species in MWW zeolite.⁶ (g) C_s -corrected iDPC STEM images of Co-MFI zeolite.⁷

Fig. R13 Optimized structure of atomically dispersed Pt species in TS-1 by DFT calculations with binding energy shown in eV.

Actions taken:

(1) The scale bar error in Fig.1e has been revised. The Fig. 1f in the initial manuscript has been moved to Supplementary Fig. 3 of the revised Supplementary Information.

(2) The Fig. 1 d has been revised in the revised manuscript.

(3) The Fig. R11 has been moved to Supplementary Fig. 4 of the revised Supplementary Information.

(4) The Fig. R13 has been moved to Supplementary Fig. 5 of the revised Supplementary Information.

(5) The evidence and explanation of the position of the Pt single atom has been supplemented in the revised manuscript:

“In addition to the Pt NPs, a considerable number of Pt single atoms were also confirmed by the Cs-corrected HAADF STEM images of Pt@HieTS-1 (Fig. 1d). Cs-corrected iDPC STEM images disclose that the Pt single atoms are located in the 5- and 6-membered rings (MR) of the MFI zeolite framework (Fig. 1e). This configuration was optimized by density functional theory (DFT) calculations (Supplementary Fig. 5). The enlarged images of three typical local structures and corresponding binding energies were listed in Fig. 1e.” (Line 11-17, Page 6, in revised manuscript)

Reference

- [1] Sun, Q. et al. Zeolite-encaged single-atom rhodium catalysts: highly-efficient hydrogen generation and shape-selective tandem hydrogenation of nitroarenes. *Angew. Chem., Int. Ed.* **58**, 18570–18576 (2019).
- [2] Deng, X. et al. Zeolite-encaged isolated platinum ions enable heterolytic dihydrogen activation and selective hydrogenations. *J. Am. Chem. Soc.* **143**, 20898–20906 (2021).
- [3] Shen, B. et al. Atomic imaging of zeolite-confined single molecules by electron microscopy. *Nature* **607**, 703–707 (2022).

- [4] Liu, L. et al. Direct imaging of atomically dispersed molybdenum that enables location of aluminum in the framework of zeolite ZSM-5. *Angew. Chem., Int. Ed.* **59**, 819–825 (2020).
- [5] Ma, Y. et al. Germanium-enriched double-four-membered-ring units inducing zeolite-confined subnanometric Pt clusters for efficient propane dehydrogenation. *Nat. Catal.* **6**, 506–518 (2023).
- [6] Liu, L., Lopez-Haro, M., Meira, D. M., Concepcion, P., Calvino, J. J., & Corma, A. Regioselective generation of single-site iridium atoms and their evolution into stabilized subnanometric iridium clusters in MWW zeolite. *Angew. Chem., Int. Ed.* **59**, 15695–15702 (2020).
- [7] Hu, Z. et al. Atomic insight into the local structure and microenvironment of isolated Co-motifs in MFI zeolite frameworks for propane dehydrogenation. *J. Am. Chem. Soc.* **144**, 12127–12137 (2022).

Comment 4. The statement “a 4.3-fold increase in reaction rate compared to unmodified catalyst” is based on a comparison with samples synthesized by the authors themselves. To demonstrate the superior performance of their catalyst in hydrogenation reactions, the authors should provide a comparative analysis with other catalysts used in this field and present this information in a table or chart for clarity and context.

Response: Thanks for your excellent suggestions. We provide a comparative analysis of other catalysts used in the field and present this information in Table R2, as follows:

Table R2 Comparison of various catalysts for the hydrogenation of benzaldehyde to

benzyl alcohol in liquid phase.

Catalyst	Solvent	Reaction conditions			Substrate conv. (%)	Product select. (%)	Ref.
		Temp. (°C)	H ₂ (atm)	Time (h)			
Pt/@-ZrO ₂ /SBA	H ₂ O	50	10	1	100	99	1
Pt/SBA-15	H ₂ O	50	10	2.5	100	99	1
Pt/ZrO	H ₂ O	50	10	2.5	29	99	1
Pd/@ZrO ₂ /AC	H ₂ O	40	7	0.5	100	98	2
Ni ₁ Fe ₁	H ₂ O	100	10	4	93	100	3
Ag-Fe ₃ O ₄ @CMC	H ₂ O	100	40	24	95	-	4
0.2Pt/MgAl ₂ O ₄	ethanol	60	10	4	100	99	5
1Pt/MgAl ₂ O ₄	ethanol	60	10	4	100	99	6
Pt/15TS	ethanol	25	40	0.75	94	99	7
Pt/TiO ₂	ethanol	25	40	0.75	85	99	7
Ni ₁ Fe ₁	ethanol	100	10	4	100	100	3
MgCoMo HT	ethanol	110	10	5	93	100	8
Pd _{NPs} /TiO ₂	ethanol	25	1	0.25	50	99	9
Pd/NGC	ethanol	30	5	1	99	42	10
Pd/AC	p-xylene	50	2	1	98	86	11
Ni-5ReO _x /TiO ₂	dioxane	120	20	0.5	63	100	12
Pd _{SA} /G	n-octane	60	7	1	99	97	13
Pt ₂ /mpgC ₃ N ₄	isopropanol	120	80	9	100	99	6
Ni/Al ₂ O ₃ -SiC	isopropanol	90	20	2	77	93	14
Cu-Pt@TMS	isopropanol	110	10	3	100	100	15
This work	H ₂ O	50	1	1	96	100	

Actions taken: Table R2 has been moved to Supplementary Table 4 in the Supplementary Information. And the corresponding explanation was made in the revised manuscript. (Line 9-12, Page 10, in revised manuscript)

Reference

- [1] Zhang, Y., Zhou, J., Wang, F., Lv, M., & Li, K. Metal-metal oxide synergistic catalysis: Pt nanoparticles anchored on mono-layer dispersed ZrO₂ in SBA-15 for high efficiency selective hydrogenation. *J. Catal.* **421**, 12–19 (2023).
- [2] Zhang, Y., Zhou, J., Li, K., & Lv, M. Synergistic catalysis of hybrid nano-structure Pd catalyst for highly efficient catalytic selective hydrogenation of benzaldehyde. *Catal. Today* **358**, 129–137 (2020).
- [3] Wang, Y. et al. Facile synthesis of Ni/Fe₃O₄ derived from layered double hydroxides with high performance in the selective hydrogenation of benzaldehyde and furfural. *Mol. Catal.* **528**, 112505 (2022).
- [4] Li, A. Y., Kaushik, M., Li, C. J., & Moores, A. Microwave-assisted synthesis of magnetic carboxymethyl cellulose-embedded Ag-Fe₃O₄ nanocatalysts for selective carbonyl hydrogenation. *ACS Sustainable Chem. Eng.* **4**, 965–973 (2016).
- [5] Yan, F. et al. Effect of the degree of dispersion of Pt over MgAl₂O₄ on the catalytic hydrogenation of benzaldehyde. *Chin. J. Catal.* **38**, 1613–1620 (2017).
- [6] Tian, S. et al. Dual-atom Pt heterogeneous catalyst with excellent catalytic performances for the selective hydrogenation and epoxidation. *Nat. Commun.* **12**, 3181 (2021).
- [7] Li, X. et al. Pt nanoparticles supported on highly dispersed TiO₂ coated on SBA-15 as an efficient and recyclable catalyst for liquid-phase hydrogenation. *J. Catal.* **300**, 9–19 (2013).
- [8] Neethu, P. P., Venkatachalam, G., Venkatesha, N. J., Joseph, D., & Sakthivel, A. Cobalt-based hydrotalcite: a potential non-noble metal-based heterogeneous

- catalyst for selective hydrogenation of aromatic aldehydes. *Ind. Eng. Chem. Res.* **62**, 4976–4986 (2023).
- [9] Kuai, L. et al. Titania supported synergistic palladium single atoms and nanoparticles for room temperature ketone and aldehydes hydrogenation. *Nat. Commun.* **11**, 48 (2020).
- [10] Mironenko, R. M. et al. Liquid-phase hydrogenation of benzaldehyde over Pd-Ru/C catalysts: synergistic effect between supported metals. *Catal. Today* **278**, 2–9 (2017).
- [11] Cattaneo, S. et al. Discovering the role of substrate in aldehyde hydrogenation. *J. Catal.* **399**, 162–169 (2021).
- [12] Lin, W. et al. Surface synergetic effects of Ni-ReO_x for promoting the mild hydrogenation of furfural to tetrahydrofurfuryl alcohol. *ACS Catal.* **13**, 11256–11267 (2023).
- [13] Yang, L. et al. Palladium single-atom catalysts synthesized by a gas-assisted redispersion strategy for efficient benzaldehyde hydrogenation. *Chem. Commun.* **59**, 5693–5696 (2023).
- [14] Li, K., Jiao, Y., Yang, Z., & Zhang, J. A comparative study of Ni/Al₂O₃–SiC foam catalysts and powder catalysts for the liquid-phase hydrogenation of benzaldehyde. *J. Mater. Sci. Technol.* **35**, 159–167 (2019).
- [15] Wang, S. et al. Ultrahigh Selective Hydrogenation of Furfural Enabled by Modularizing Hydrogen Dissociation and Substrate Activation. *ACS Catal.* **13**, 8720–8730 (2023).

Comment 5. In the first paragraph of results and discussions section, the sentence “Then, rendering the external surface of the zeolite hydrophobic by appending organosilanes, namely Pt@HieTS-1-C_x (C_x represents the organic substituent of the silane sheath).” appears to be incomplete. The authors should check the manuscript carefully.

Response: Thanks for your carefully checking the language of our manuscript. We feel really sorry that we did not provide a clear description of this point in our initial manuscript. Accordingly, this point has been corrected in our revised manuscript. We also have revised the whole manuscript carefully and tried to avoid any grammatical errors and badly worded/constructed sentences.

Actions taken:

“Then, the external-surface hydrophobization of zeolite is obtained by modification with organosilanes (Pt@HieTS-1-C_x, C_x represents the organic substituent of the silane sheath).” (Line 14-16, Page 5, in revised manuscript)

Response to Reviewer #4

The authors reported a zeolite nanoreactor with joint gas-liquid-solid interface. The micro environment of the catalytic site was modified by surface hydrophobization to enhance the concentration of H₂ and aldehydes/ketones substrate. Some interesting results and mechanism understanding are obtained.

Response: We thank the reviewer for thoughtful comments, which are valuable in improving the quality of our manuscript. As described below, we have made a detailed explanation and performed additional experiments to address all the comments: (1) The innovation of this research was highlighted in comparison with the previously reported works. (2) The reaction evaluation and the quantitative analysis were further checked. (3) A more comprehensive explanation of the contact angle test was provided. (4) The temperature sensitivity of wettability on hydrogenation performance was investigated. (5) The influence of H₂O molecules has been considered in MD simulations. (6) More experimental details were supplemented.

Comment 1. However, the concept of enhancing the performances of heterogeneous catalysts by constructing hydrophobic or hydrophilic surface has been reported in several papers, for example in ref. 12-17 as mentioned by the authors. In particular, very similar ideas were reported in refs. 15 and 16 (*References 1*: Mi et al., JACS, 2017, 139, 10441; *References 2*: Li et al., AM Interfaces, 2018, 5, 1801259). This decreases the novelty of the present work. It is more suitable to more specific journals on catalysis. In addition, the following issues should be adequately addressed.

Response: Thank you for your insightful comments.

In *Reference 1* mentioned by the reviewer #4, Lei Jiang group reported a nanochannel reactor with joint gas-solid-liquid interfaces and controlled wettability. A porous anodic alumina nanochannel membrane with different wettability was used for glucose oxidase immobilization, which contacts with glucose aqueous solution on one side, while the other side gets in touch with the gas phase directly. As a result, the O₂ could participate in the enzymatic reaction directly from gas phase through the proposed nanochannels. In *Reference 2*, Pd nanoparticles were loaded on graphene aerogel with different degrees of aerophilic/aerophobic surfaces. By creating supraaerophilic surface to increase H₂ concentration on catalyst surface, the hydrogenation reaction rate was significantly enhanced in aqueous solution.

The common denominator of these references is to solve the gas-deficit problem in the gas-liquid-solid three-phase reaction by regulating the wettability of the catalyst surface. As we all know, a catalyst with excellent catalytic performance requires mass transfer matching during the reaction. The mass transfer in gas-liquid-solid three-phase hydrogenation reactions involves the diffusion of H₂, substrates and products. However, the simultaneous regulation of mass transfer of H₂, substrates and products in hydrogenation has been rarely explored.

In this manuscript, we present a zeolite nanoreactor with controlled wettability for efficient mass transfer of H₂, reactant and product molecules at gas-liquid-solid three-phase interface. Specifically, the Pt active sites are encapsulated within zeolite crystals, followed by modifying the external zeolite surface with organosilanes. This zeolite

nanoreactor with aerophilic/hydrophobic property can promote the diffusion of H₂ and the mass transfer of reactant/product molecules. Specifically, in aqueous solutions, the gaseous H₂ molecules can rapidly diffuse into the zeolite channels. Simultaneously, the silane sheath with lipophilicity nature promotes the enrichment of the aldehydes/ketones on the catalyst. Finally, the silane sheath also facilitates the hydrophilia products of alcohol rediffusion back to the aqueous phase, accelerating the solid-liquid mass transfer. By modifying the wettability of the catalyst, the hydrogenation of aldehydes/ketones can be operated in water at ambient H₂ pressure, resulting in a noteworthy turnover frequency up to 92.3 h⁻¹ and a 4.3-fold increase in reaction rate compared to the unmodified catalyst. This work innovatively proposes a method for simultaneously regulating the mass transfer of H₂, reactant and product molecules in gas-liquid-solid three-phase hydrogenation reactions.

Comment 2. The critical issue is the reaction evaluation and the quantitative analysis. The authors only described that “liquid products were analyzed by a gas chromatography spectrometry”, how did the author calculate the conversion? Is it based on the relative peak area of benzaldehyde and benzyl alcohol? Did the authors use internal standard chemical and calculate the carbon balance?

Response: Thank you for your specialized comments.

(1) After the reaction, the organic phase was extracted with ethyl acetate. The conversion and yield were quantitatively analyzed by gas chromatography spectrometry. External standard method was used for the product quantification in the current study.

The concentration of benzaldehyde and benzyl alcohol was calculated according to their corresponding standard curves.

Actions taken: The details of reaction evaluation and quantitative analysis have been listed in the revised manuscript. As follows:

“The liquid-phase hydrogenation of aldehydes/ketones was performed in a glass round-bottom flask reactor. The substrate was dispersed in water and uniformly distributed by ultrasound, followed by the addition of catalyst. The air in reactor is removed by vacuuming and then H₂ (1 atm) was injected. The reaction was initiated by heating to the designated temperature while stirring at 900 rpm. After the reactions, the organic phase was extracted from the aqueous phase using ethyl acetate, with an extraction-to-solvent ratio of 2:1 (v/v). The conversion and yield were analyzed by a gas chromatography spectrometry (GC-2010 Plus, MXT-1 column). External standard method was used for the product quantification in the current study. The carbon balance of all examined catalysts was in the range of 96 to 99%. The catalytic reaction data were calculated based on the following formulas:

$$\text{Conversion (\%)} = \left(1 - \frac{\text{molar amount of substrate after reaction}}{\text{initial molar amount of substrate fed}}\right) \times 100\%$$

$$\text{Yield (\%)} = \frac{\text{molar amount of one product}}{\text{initial molar amount of substrate fed}} \times 100\% \text{ (Line 9-22, Page 18 and Line 1-3,}$$

Page 19, in revised manuscript)

(2) The carbon balance has been calculated and carbon balance of all examined catalysts was in the range of 96 to 99%.

Actions taken: The carbon balances for some important reactions have been summarized in Table R3.

Table R3 Catalytic hydrogenation of benzaldehyde with different catalysts.^a

Entry	Catalysts	Solvent	H ₂ (atm)	Time (h)	Yield of benzyl alcohol (%)	Carbon balance (%)
1	Pt@HieTS-1	H ₂ O	1	8	94	97
2	Pt@HieTS-1-C ₃	H ₂ O	1	2.5	99	99
3	Pt@HieTS-1-C ₆	H ₂ O	1	3.5	99	97
4	Pt@HieTS-1-C ₁₆	H ₂ O	1	4.5	98	96
5	Pt@HieTS-1	H ₂ O	10	0.5	22	97
6	Pt@HieTS-1-C ₃	H ₂ O	1	0.5	26	99

^a Reaction conditions: benzaldehyde (0.47 mmol), Pt dosage (2.1×10^{-3} mmol), solvent (5 mL), 50 °C.

Actions taken: Table R3 has been moved to Supplementary Table 5 in the Supplementary Information. And the corresponding explanation was made in the revised manuscript. (Line 13-15, Page 10, in revised manuscript)

Comment 3. It should be noted that the water solubility of benzaldehyde (BA) is extremely low and the concentration of BA (1%) used in the current work would lead to the formation of oil-water phase, which could greatly affect the accuracy of quantitative analysis.

Response: Thanks for your comments. Before starting the reaction, the benzaldehyde/water mixture was dispersed by ultrasound, forming a uniform emulsion.

During the reaction process, the emulsion has been subjected to intense magnetic agitation (900rpm) to ensure the uniform dispersion of benzaldehyde in the aqueous phase. After the reaction, the organic phase was separated from the aqueous phase by extraction with ethyl acetate. Separating the organic phase from the aqueous phase by extraction is an effective method for the quantitative analysis in the hydrogenation reaction.^{1,2}

Reference

- [1]. Huang, J., Cheng, F., Binks, B. P., & Yang, H. pH-responsive gas-water-solid interface for multiphase catalysis. *J. Am. Chem. Soc.* **137**, 15015–15025 (2015).
- [2]. Vjunov, A., Derewinski, M. A., Fulton, J. L., Camaioni, D. M., & Lercher, J. A. Impact of zeolite aging in hot liquid water on activity for acid-catalyzed dehydration of alcohols. *J. Am. Chem. Soc.* **137**, 10374–10382 (2015).

Comment 4. The catalyst weight for each reaction was about 58 mg, and only 50 mg BA reactant was added for each reaction. It should be noted that the BA reactant (lipophilic) can be adsorbed on the catalyst especially on the lipophilic Pt@HieTS-1-C_x, which was also investigated by the authors in Supplementary Figure 13. It is possible that the BA in the oil-water phase is adsorbed on the catalyst but not completely dissolved in the water phase (only BA dissolved in water phase can be detected by GC). The above issue could lead to the overestimate of BA conversion.

Response: Thank you for your comments. After verification, the adsorption of lipophilic benzaldehyde substrate on the lipophilic Pt@HieTS-1-C_x can be separated

by extraction and does not affect the calculation of conversion. Specifically, at the end of the reaction, both the product and unreacted substrate are fully extracted from the aqueous phase using ethyl acetate (the unreacted substrates include dispersed in the aqueous phase and adsorbed on the catalyst). It has been confirmed by adsorption experiments that both the substrate dispersed in water and adsorbed on the lipophilic Pt@HieTS-1-C_x can be fully extracted into ethyl acetate. As shown in Fig. R6, we repeated the adsorption experiment several times and calculated the carbon balance of the adsorption experiment to be about 95~97%. This means that almost all adsorbed benzaldehyde substrates can be extracted into ethyl acetate. Therefore, method for the calculation of conversion in the present work is accurate and reliable.

Comment 5. Experimental details for the reaction evaluation and the quantitative analysis should be fully addressed. Carbon balance for each reaction should be carefully calculated. To enhance the reliability of the data, error bars (based on several repeated experiment) for some important reaction data is suggested to be added.

Response: Thanks for your suggestion.

(1) We have provided experimental details for the reaction evaluation and quantitative analysis in the Supplementary Information.

Actions taken: The details of reaction evaluation and quantitative analysis have been added. (Line 10-22, Page 18 and Line 1-4, Page 19, in revised manuscript)

(2) We have carefully calculated the carbon balance, including catalytic performance evaluation and adsorption experiments. The carbon balance of all examined catalysts

was in the range of 96 to 99%.

Actions taken: The carbon balances for the related reactions or adsorption experiments have been corresponding annotated. (Supplementary Table 5 and Supplementary Fig. 21) (Line 13-15, Page 10, in revised manuscript)

(3) We have conducted several repeated experiments on some important reactions and presented the error bars in the reaction results.

Updated Fig. 4 (a) The evolution of benzaldehyde conversion with the reaction time on various catalysts. (b) The kinetic plots of the catalysts, calculated at the low conversion of benzaldehyde. (c) The corresponding rate constant value of various catalysts. (d) The TOF values of the catalysts.

Actions taken: Fig. 4 a-d have been revised and the corresponding explanation was made in the revised manuscript.

Comment 6. The details of the contact angle measurement should be provided. Is the

catalyst in Fig. 3b, e compressed into slice? If so, how the authors assure it has similar wetting properties with the particulate catalysts. In addition, scale bars should be added in Fig. 3b, e and related supplementary figures.

Response: Thanks for your thoughtful comments.

(1) About the details of the contact angle measurement.

The details of the contact angle measurements have been added in revised manuscript. The sessile drop technique was used to measure the contact angle in this work. For powder samples, this method requires the compaction of powder into a wafer by applying a high pressure. The contact angle measurements have been listed as follows:

“The contact angles were measured using a contact angle system (OCA 20, Dataphysics) at ambient temperature, with the probe liquid being 10 μ L. For sample preparation, 50 mg of the catalyst was compressed into a wafer with a diameter of 13 mm and a smooth surface under a pressure of 10 MPa. Contact angle images were taken after the application of the liquid droplet on the surface of samples (or the application of H_2 gas bubble on the surface of samples underwater).” (Line 4-9, Page 18, in revised manuscript)

(2) About the wetting properties of powder and wafer samples.

The sessile drop technique used in this work is one of the most common methods to determine the wettability of powder catalysts.¹⁻⁴ For powder samples, this method requires the compaction of powder into a wafer by applying a high pressure. The simplicity and reproducibility of this technique make it advantageous as compared to

other tedious techniques. However, there are limitations to this approach,⁵⁻⁶ such as compacting pressure affects surface properties of powder, surface structure leads to drop penetration and surface roughness leads to air trapping, resulting in only the apparent contact angle. In the contact angle measurements of this work, all samples were done under a uniform operating procedure, and the surface of the wafer is smooth. After the solid powder pressed into wafer, it does not damage the microstructure of the catalyst (Fig. R14). In this regard, this method enables a qualitative comparison of the wettability between different samples, reflecting the wettability trend of particulate catalysts.

Since the contact angle test only displays the hydrophobic property in macroscopic manner, confocal laser scanning microscopy (CLSM) technique was further carried out to investigate the hydrophobic property at micro-nano level (i.e. in the particulate state). Water-soluble fluorescent dye was selected as the fluorescent probe molecule. In this regard, fluorescent dye can only lighten the area where water is present. Taken Pt@HieTS-1-C₃ and Pt@HieTS-1 as contrast examples, the schematic diagram and the CLSM images are shown in Figure 3c. It can be seen that fluorescent dye can lighten the whole Pt@HieTS-1 body, whereas very weak fluorescence signal is observed on Pt@HieTS-1-C₃ body, indicating that water cannot penetrate into the channels of Pt@HieTS-1-C₃. Therefore, the wettability of the catalyst in the particle state was understood from the microscopic scale.

Fig. R14 SEM images of Pt@HieTS-1-C₃ with (a) particle state and (b) wafer state.

(3) About the scale bars of contact angle images.

We have added scale bars to the images involving contact angle measurements, as follows:

Updated Fig. 3 (b) The contact angles of water with Pt@HieTS-1, Pt@HieTS-1-C₃, Pt@HieTS-1-C₆ and Pt@HieTS-1-C₁₆.

Updated Fig. 3 (e) The contact angles of H₂ gas-bubble with catalyst under water.

Updated Supplementary Fig. 8 Contact angles of benzaldehyde with (a) Pt@HieTS-1, (b)Pt@HieTS-1-C₃, (c) Pt@HieTS-1-C₆ and (d) Pt@HieTS-1-C₁₆.

Reference

- [1]. Mi, L. et al. Boosting gas involved reactions at nanochannel reactor with joint gas–solid–liquid interfaces and controlled wettability. *J. Am. Chem. Soc.* **139**, 10441–10446 (2017).
- [2]. Fang, M. et al. Hydrophobic, ultrastable Cu^{δ+} for robust CO₂ electroreduction to C₂ products at ampere-current levels. *J. Am. Chem. Soc.* **145,20**, 11323–11332 (2023).
- [3]. Xie, L. et al. WS₂ moiré superlattices derived from mechanical flexibility for hydrogen evolution reaction. *Nat. Commun.* **12**, 5070 (2021).
- [4]. Shi, R. et al. Efficient wettability-controlled electroreduction of CO₂ to CO at Au/C interfaces. *Nat. Commun.* **11**, 3028 (2020).
- [5]. Alghunaim, A., Kirdponpattara, S., & Newby, B. M. Z. Techniques for determining contact angle and wettability of powders. *Powder Technol.* **287**, 201–215 (2016).
- [6]. Nowak, E., Combes, G., Stitt, E. H., & Pacey, A. W. A comparison of contact angle measurement techniques applied to highly porous catalyst supports. *Powder Technol.* **233**, 52–64 (2013).

Comment 7. The authors only investigate the catalyst at 50 °C. The surface wetting property may be sensitive to the temperature. In what temperature range does the catalyst maintain this high performance?

Response: Thanks for your thoughtful comments. According to the reviewer's suggestion, we investigated the temperature-dependent sensitivity of surface wettability. As shown in Fig. R13, the reactivity of Pt@HieTS-1 and Pt@HieTS-1-C₃ exhibited an ascending trend concomitant with elevated temperatures. Within the temperature range of 50-110 °C, the catalytic performance of Pt@HieTS-1-C₃ catalyst modified by surface wettability markedly surpasses that of unmodified catalyst.

Fig. R15 The effect of the reaction temperature on benzaldehyde hydrogenation. Reaction condition: benzaldehyde (0.47 mmol), Pt dosage (2.1×10^{-3} mmol), water (5 mL), 1 atm H₂, 0.5 h.

Actions taken: Fig. R15 has been moved to Supplementary Fig. 16 of the revised Supplementary Information. The corresponding explanation has been added in the revised manuscript:

“The temperature-dependent sensitivity of surface wettability was also investigated. As

shown in Supplementary Fig. 16, within the temperature range of 50-110 °C, the catalytic performance of Pt@HieTS-1-C₃ catalyst markedly surpasses that of unmodified catalyst.” (Line 17-21, Page 12, in revised manuscript)

Comment 8. The authors carried MD without considering the effect water molecules, which is different from the real situation. This gap should be addressed.

Response: Thanks for your comments. In molecular dynamics (MD) simulations, we have fully considered the effect of water molecules.

(1) In the simulation of benzaldehyde molecular diffusion, 400 molecules of benzaldehyde molecules and 2640 molecules of water molecules (solvent) were both located at the left of TS-1 and TS-1-C₃ with the distance about 0.5 nm (Fig. R16). After simulating diffusion within 1 ns, it is obvious that water molecules can easily diffuse into the TS-1 channels (Fig. 6a, b). In contrast, water molecules cannot diffuse into hydrophobic TS-1-C₃ channels.

(2) In the simulation diffusion of H₂ molecular, 400 molecules of H₂ molecules and 2640 molecules of water molecules (solvent) were both located at the left of TS-1 and TS-1-C₃ with the distance about 0.5 nm (Fig. R17). As shown in Fig. 6c, d, after simulating diffusion within 1 ns, water molecules can easily diffuse into the TS-1 channel but not into the TS-1-C₃ channel.

Fig. R16 The snapshot of initial state on the benzaldehyde diffusion.

Fig. 6 The snapshot of the penetration of benzaldehyde into (a) TS-1 and (b) TS-1-C₃ in water solution.

Fig. R17 The snapshot of initial state on the H₂ diffusion.

Fig. 6 The snapshot of the penetration of H_2 into (c) TS-1 and (d) TS-1- C_3 in water solution.

Actions taken:

- (1) Fig. R16 has been moved to Supplementary Fig. 19.
- (2) Fig. R17 has been moved to Supplementary Fig. 20.
- (3) The corresponding statement has been revised in the manuscript:

“In the initial state, 400 molecules of benzaldehyde (or H_2) molecules and 2640 molecules of water molecules (solvent) were located at the left of TS-1 and TS-1- C_3 with the distance about 0.5 nm (Supplementary Fig. 18-20).” (Line 1-4, Page 14, in revised manuscript)

Comment 9. How did the author calculate the adsorption capacity (Supplementary Figure 13)? How was this adsorption experiment carried out? Experimental details must be addressed.

Response: Thanks for your suggestion. The details of the adsorption experiments have been supplemented.

Actions taken: The revisions are as follows:

“In typical procedure,¹⁻³ the adsorption experiment was carried out in a 15 mL glass bottle containing 5 mL water solution with 0.47 mmol benzaldehyde (or benzyl alcohol). Benzaldehyde (or benzyl alcohol) was dispersed evenly in water by ultrasound, and then as-prepared catalyst was added. The glass bottle equipped with magnetic stirring (900rpm) was heated to 50 °C, and the adsorption experiment lasted for 1h. Subsequently, the catalyst and the mixture were separated by filtration. The aqueous solution was extracted with ethyl acetate to ensure that the unadsorbed benzaldehyde (or benzyl alcohol) is dissolved into the ethyl acetate. The catalyst was extracted using ethyl acetate, guaranteeing the sufficiently desorption of benzaldehyde (or benzyl alcohol) adsorbed on the catalyst into the ethyl acetate phase. After that, quantitative analyses were conducted by GC. The dosage of benzaldehyde (or benzyl alcohol) adsorption was calculated by the difference of concentration before and after the adsorption experiment. The carbon balance of the adsorption experiments approximated 95~97%.” (Page 4 and Page 5, in revised supplementary information)

Reference

- [1]. Li, R. et al. Hydrodeoxygenation of phenolic compounds and raw lignin-oil over bimetallic RuNi catalyst: an experimental and modeling study focusing on adsorption properties. *Fuel* **281**, 118758 (2020).
- [2]. Guo, Y. et al. One-pot synthesis of sulfur doped activated carbon as a superior metal-free catalyst for the adsorption and catalytic oxidation of aqueous organics. *J. Mater. Chem.* **6**, 4055–4067 (2018).
- [3]. Shu, R. et al. Enhanced adsorption properties of bimetallic RuCo catalyst for the

hydrodeoxygenation of phenolic compounds and raw lignin-oil. *Chem. Eng. Sci.*

227, 115920 (2020).

REVIEWERS' COMMENTS

Reviewer #1 (Remarks to the Author):

I am satisfied with the revision made by the authors.

Reviewer #2 (Remarks to the Author):

The authors have satisfactorily addressed the reviewers' comments. Currently I think that this work is acceptable for publication.

Reviewer #3 (Remarks to the Author):

The authors have addressed the concerns I previously raised by providing iDPC-STEM and HAADF-STEM images of the Pt@HieTS-1 zeolite. They have demonstrated the presence of Pt atoms by showing HAADF-STEM images of the material in comparison to HieTS-1 zeolite without Pt loading. Additionally, they have furnished comparisons with other studies in the relevant field as requested. Upon careful examination of the revised manuscript, I find that the authors have effectively resolved the issues I highlighted and corrected the original inaccuracies, thereby substantiating the conclusions of their paper. Consequently, I believe this article is now suitable for publication in Nature Communications. Followings are my comments on each response from the reviewers.

Comment 1.

The authors clarified how their synthesis method is distinct from others, specifically noting that the Pt active sites are in-situ encapsulated within the HieTS-1 zeolite framework.

Comment 2.

The authors have amended the previously unclear images in the manuscript and provided clearer visualizations. In these new images, they combined HAADF-STEM, iDPC-STEM, and DFT analyses to annotate the locations of the Pt atoms.

Comment 3.

The authors have firstly corrected the scale errors in the original images. Additionally, they presented HAADF-STEM images of both Pt-loaded and unladen HieTS-1 zeolite. I believe this comparative approach convincingly demonstrates the existence of Pt atoms. Moreover, the use of DFT to elucidate the anticipated positions of Pt atoms within the HieTS-1 zeolite framework aligns well with their revised images, affirming the accuracy of the authors' conclusions.

Comment 4.

The authors have included a comparative analysis with other studies in the same field as suggested. In this comparison, the performance of Pt@HieTS-1 zeolite appears to show certain improvements over other studies.

Comment 5.

The authors have rectified the language inaccuracies previously present in the manuscript.

Reviewer #4 (Remarks to the Author):

The authors have adequately addressed the issues I raised. The manuscript is now acceptable for publication in Nature Communications.

RESPONSE TO REVIEWERS' COMMENTS

Response to Reviewer #1:

Comment: I am satisfied with the revision made by the authors.

Response: Thank you for your approval and positive feedback on our revised manuscript. We are pleased to hear that the changes have met your satisfaction. We appreciate your guidance and support throughout the review process, which has significantly contributed to improving our work.

Response to Reviewer #2:

Comment: The authors have satisfactorily addressed the reviewers' comments. Currently I think that this work is acceptable for publication.

Response: Thank you for your positive feedback and for considering our manuscript suitable for publication. We are grateful for the opportunity to address the reviewer #2' comments, which have undoubtedly strengthened our work.

Response to Reviewer #3:

Comment: The authors have addressed the concerns I previously raised by providing iDPC-STEM and HAADF-STEM images of the Pt@HieTS-1 zeolite. They have demonstrated the presence of Pt atoms by showing HAADF-STEM images of the material in comparison to HieTS-1 zeolite without Pt loading. Additionally, they have furnished comparisons with other studies in the relevant field as requested. Upon careful examination of the revised manuscript, I find that the authors have effectively resolved the issues I highlighted and corrected the original inaccuracies, thereby

substantiating the conclusions of their paper. Consequently, I believe this article is now suitable for publication in Nature Communications.

Response: We sincerely appreciate your thorough review and constructive comments on our manuscript. We are glad that the additional iDPC-STEM and HAADF-STEM images, along with the expanded comparisons to relevant studies, have successfully addressed your concerns and clarified the presence of Pt atoms in our Pt@HieTS-1 zeolite. Your feedback has been invaluable in enhancing the accuracy and depth of our work. We are honored by your assessment that our article is now suitable for publication in Nature Communications.

Response to Reviewer #4:

Comment: The authors have adequately addressed the issues I raised. The manuscript is now acceptable for publication in Nature Communications.

Response: Thank you very much for your constructive feedback and for acknowledging the revisions made to our manuscript. We are pleased to hear that it is now deemed acceptable for publication in Nature Communications. We appreciate your guidance and the opportunity to enhance our work through the review process.